# StTCTP Positively Regulates *StSN2* to Enhance Drought Stress Tolerance in Potato by Scavenging Reactive Oxygen Species

**DOI:** 10.3390/ijms26062796

**Published:** 2025-03-20

**Authors:** Shifeng Liu, Feng Zhang, Haojie Feng, Xiyao Wang, Qiang Wang, Xianjun Lai, Lang Yan

**Affiliations:** 1Panxi Crops Research and Utilization Key Laboratory of Sichuan Province, College of Agricultural Science, Xichang University, Liangshan 615300, China; 13838657723@163.com; 2Potato Research and Development Center, College of Agronomy, Sichuan Agricultural University, Chengdu 611130, China; zhangfeng_95@126.com (F.Z.); fenghaojie@stu.sicau.edu.cn (H.F.); wangxy@sicau.edu.cn (X.W.); qwang@sicau.edu.cn (Q.W.)

**Keywords:** *StTCTP*, *StSN2*, drought tolerance, antioxidant enzyme activity, ROS

## Abstract

Drought is a negative agronomic effect that can lead to an increase in reactive oxygen species (ROS) levels. Excessive drought can severely alter cell membrane fluidity and permeability, significantly reducing cell viability. The *Gibberellic acid-stimulated Arabidopsis* (Snakin/GASA) gene family has an important role as antioxidants in inhibiting the accumulation of ROS and improving crop drought resistance. However, the regulatory mechanism of potato *StSnakin-2* (StSN2) in response to drought, along with how *StSN2* expression is regulated, is not well understood. In this study, we found that *StSN2* was induced by drought. Overexpression of *StSN2* significantly increased drought tolerance, whereas silencing *StSN2* increased sensitivity to drought. Overexpression of *StSN2* resulted in higher antioxidant enzyme (superoxide dismutase (SOD), catalase (CAT), and peroxidase (POD)) activity, and lowered hydrogen peroxide (H_2_O_2_) and malondialdehyde (MDA) accumulation during drought stress. Also, overexpression of *StSN2* increased the relative water content (RWC) of leaves and reduced the water loss in leaves. We screened the upstream regulatory protein translation-controlled tumor protein (StTCTP) of *StSN2* through DNA pull-down combined with mass spectrometry. Yeast one-hybrid (YIH), electrophoretic mobility shift assay (EMSA), and luciferase reporting assay (LUC) indicated that StTCTP binds the *StSN2* promoter. Like *StSN2*, *StTCTP* was highly expressed in response to drought. Overexpression of *StTCTP* increased the photosynthetic rate and CAT enzyme activity, and lowered H_2_O_2_ and MDA accumulation during drought. Meanwhile, overexpression of *StTCTP* increased leaf RWC and reduced water loss. Our research strongly suggested that *StSN2* effectively cleared ROS and significantly boosted the drought resistance of potatoes. Furthermore, as a transcriptional activator of *StSN2*, *StTCTP*, much like *StSN2*, also enhanced the potato’s drought tolerance. The results provided a foundation for the further study of *StSN2* regulatory mechanisms under drought stress.

## 1. Introduction

Water is the source of life and one of the main components of cells [1]. It is an indispensable medium for all physiological activities and biochemical reactions within cells. The drought-induced water deficit can have a profoundly negative impact on crop development and growth [2]. Extended periods of drought can cause reduced yield or complete crop failure [3]. With global warming, droughts will become more severe in the future. Potato (*Solanum tuberosum* L.) is the fourth food on earth and plays a crucial role in food security and animal husbandry. The nutritional components of potatoes contain antioxidants, which have a significant impact on human health [4]. Although potatoes have been traditionally regarded as drought-resistant crops, the increasing intensity of drought, triggered by global greenhouse effects and water scarcity, has had a significant impact on both the yield and quality of potatoes. Improving the drought tolerance of potato varieties has become a major focus of potato cultivar improvement efforts. These efforts to enhance drought resistance in potatoes require the discovery and utilization of drought-resistant genes, along with a deeper understanding of how these genes are regulated.

Adverse environments can indeed induce oxidative stress responses in plants and lead to the accumulation of a large amount of ROS [5]. Plants produce a large amount of ROS during drought, which can cause oxidative damage to nucleic acids, proteins, and cell membranes in cells [6,7]. To prevent oxidative stress from causing toxicity to cells, plant cells have developed multiple antioxidant mechanisms to maintain the homeostasis of intracellular ROS levels. The *Snakin/GASA* gene family is a class of small molecular antioxidant proteins that are widely distributed in diverse plant species [8]. These proteins contain a redox active site and are thought to play a role in redox regulation. Overexpression of *GIP2* (Snakin-2) in *Petunia hybrida* led to a reduced accumulation of H_2_O_2_ in leaves following wounding [9]. Overexpression of *AtGASA4* and *AtGASA14* in *Arabidopsis* resulted in a reduced accumulation of ROS and enhanced stress tolerance [10,11]. *StSN1* plays a crucial role in maintaining redox balance. Silencing *StSN1* leads to an increase in ROS and ascorbic acid accumulation in leaves [12]. StSN2 belongs to the Snakin/GASA family of proteins. In a recent study by our group, we found that overexpression of *StSN2* inhibits germination by reducing the H_2_O_2_ content of tubers [13]. StSN2 associates with cytoplasmic glyceraldehyde-3-phosphate dehydrogenase (StGAPC) and inhibits the oxidative activity of StGAPC, which inhibits bud growth [14]. *StSN2* is a key negative regulatory factor of ROS accumulation in potatoes. However, its function under drought stress and the response mechanism of drought on *StSN2* have not been fully demonstrated.

TCTP is a highly conserved multifunctional protein that plays an important role in plant growth, development, and stress response [15]. *AtTCTP* is a positive regulator of drought, which can reduce leaf transpiration rates through ABA-induced stomatal closure, thereby improving drought tolerance in *Arabidopsis* [16]. Fatty acids are the main component of cell membranes, and drought can lead to an increase in fatty acid unsaturation and membrane permeability [17]. Overexpression of *SlTCTP* in tobacco can induce the expression of fatty acid metabolism genes *Omega-3* and *Delta-8*, maintain cell membrane permeability, and enhance plant drought resistance [18]. Salicylic acid (SA) is an important plant hormone that can regulate plant growth and development and widely participates in various stress responses [19]. Wheat *TCTP* regulates the SA signaling pathway to reduce H_2_O_2_ accumulation and enhance drought resistance [20]. Under salt stress, the expression level of *TCTP* in *Jatropha curcas* increases [21]. During low temperature and drought stress, the expression of *TCTP* in *Hevea brasiliensis* increases [22]. In summary, plant TCTP protein plays a positive role in responding to abiotic stress, but the specific mechanism of action is not yet clear. As a transcription factor, TCTP could directly bind to the Sf1 site of the octamer binding transcription factor 4 (Oct4) promoter and activate *Oct4* transcription in *Xenopus oocytes* [23]. However, in mice, TCTP binds the Sf1 site of the *Oct4* promoter and reduces the transcription level of *Oct4* pluripotent cells [24], indicating that TCTP may have different functions in different species. In addition, the transcriptional activation function of TCTP has only been validated in animal cells and has not been studied in plants yet. Therefore, the research on the transcriptional activation function of TCTP in potato will help further improve the molecular regulatory network of plant *TCTP* and provide solid theoretical support for screening potato varieties with stress resistance.

Potatoes, a key crop with high nutritional and economic value, often face drought stress during production [25]. The *Snakin/GASA* gene family, involved in plant stress responses [26], includes the drought-responsive gene *StSN2*, which is highly expressed in potato leaves under drought. *StSN2* reduces ROS accumulation by boosting antioxidant enzyme activity, thus protecting cells from oxidative damage. Additionally, StTCTP has been identified as interacting with the *StSN2* promoter through DNA pull-down and mass spectrometry, confirmed by Y1H, EMSA, and LUC assays. Both *StSN2* and *StTCTP* are drought-inducible, revealing a new drought response pathway in potatoes. These findings uncover a novel drought response mechanism in potatoes, offering key insights for developing drought-resistant varieties. The further study of *StSN2* and *StTCTP* genes will clarify their roles in drought resistance, aiding in the breeding of resilient potato varieties and providing valuable insights for drought research in other crops.

## 2. Results

### 2.1. StSN2 Was Upregulated by Drought Stress

Given that the *Snakin/GASA* genes have a function of resisting abiotic stress [27], it is reasonable to infer that the *StSN2* gene may also be concerned with drought resistance in plants. We used the slow drought method to treat potato seedlings that had grown for 60 days. The results showed that, as soil moisture decreased, the potato seedlings grew slowly and even exhibited stagnant growth. On the 8th day, the leaves began to shrink, and, on the 12th day, the drought damage reached its maximum value (Figure 1A). We detected the expression level of *StSN2* during the same period, and its expression trend was consistent with the growth of potatoes. On the 0th and 4th days, the expression of *StSN2* increased rapidly. However, on the 8th and 12th days, the expression of *StSN2* increased even more significantly, reaching levels 6.5 times and 12.8 times higher than on day 0, respectively (Figure 1B). The spatial expression analysis revealed that *StSN2* was particularly expressed in leaves on day 12 post-drought treatment, at levels that were 12.4 and 2 times higher than those in stems and roots, respectively (Figure 1C). In summary, these results indicate that *StSN2* is a gene that responds to drought stress.

### 2.2. StSN2 Enhances Drought Tolerance in Potato

PEG-6000 is a frequently used osmotic agent capable of restricting plants’ absorption of water from soil, thereby simulating the drought stress [28]. We previously obtained overexpression and silencing strains of *StSN2* (Appendix A). To investigate whether the stable expression of *StSN2* enhanced resistance to drought, we inoculated transgenic potato shoot tips (2–3 cm) onto fresh MS plates supplemented with PEG-6000; fresh MS plates served as controls. After 4 weeks, *StSN2* overexpression lines exhibited normal growth relative to WT plants when grown on MS medium alone. However, they developed longer shoots and roots than WT plants when grown on MS medium containing 5%PEG-6000 (Figure 2A). Plant length results showed that, compared with the WT, the *StSN2* overexpression lines were 1.56- and 1.47-fold higher (Figure 2B), whereas those in the two RNAi lines were 0.84- and 0.87-fold higher; the root lengths of two *StSN2* overexpression lines were 2.1- and 3.24-fold higher than WT, whereas those of the two RNAi lines were 0.25- and 0.37-fold higher (Figure 2C). Leaf RWC is considered an important parameter for drought resistance [29]. Consistent with the drought tolerance phenotype, the *OE-StSN2* lines exhibited significantly higher values (>50%) than those of the *RNAi-StSN2* (about 35%) (Figure 2E). Similarly to RWC, the water loss rate of the detached leaves from the *OE-StSN2* lines was significantly lower than WT and *RNAi-StSN2* (Figure 2D). From the above, it can be seen that *OE-StSN2* has a higher tolerance to drought than WT and *RNAi-StSN2*.

H_2_O_2_ quantification and 3, 30-diaminobenzidine (DAB) staining were used to assess oxidative damage caused by drought stress in the overexpression and RNAi lines. Overexpression of *StSN2* resulted in reduced H_2_O_2_ accumulation, while silencing *StSN2* resulted in increased H_2_O_2_ accumulation under drought stress conditions. There was no significant difference in H_2_O_2_ accumulation among the overexpression, RNAi, and WT lines under non-stress conditions (Figure 3A,B). Antioxidant enzymes are the main intracellular ROS scavenger, they can effectively neutralize and reduce the damage caused by oxidative stress to cells [30]. We measured the activity of POD, SOD, and CAT, along with the content of MDA. Results showed that there was no significant difference in the activity of POD, SOD, and CAT, along with the content of MDA between the overexpression, RNAi, and WT lines under non-stress conditions. Under drought stress conditions, *StSN2* promoted an increase in the activity of POD, SOD, and CAT enzymes, while reducing the content of MDA (Figure 3C–F). These results indicate that *StSN2* can activate the ROS scavenging system to enhance drought tolerance in potatoes.

### 2.3. StTCTP Is an Upstream Regulatory Factor of StSN2

In order to identify regulatory factors that specifically bind to the *StSN2* promoter, we identified the *StSN2* promoter sequence using the PGSC database (Appendix A). We successfully cloned the *StSN2* promoter sequence (942 bp) using PCR technology and constructed a GUS fusion expression vector. A fluorometric analysis of β-glucuronidase (GUS) activity showed that the cloned *StSN2* promoter could drive GUS gene expression (Appendix A). We used the biotin/avidin binding system (DBAS) to isolate specific regulators that are bound to the *StSN2* promoter (Appendix A). StTCTP was identified as a potential *StSN2* promoter-binding protein (Appendix A). We validated this interaction using a dual luciferase assay in which the *StSN2* promoter was placed upstream of luciferase (Figure 4A) and coexpressed with StTCTP in *N. benthamiana* leaves. A gradual increase in luminescence was observed when the *StSN2pro-LUC* was co-transformed with the *35Spro-StTCTP* relative to the empty effector vector (Figure 4C). LUC activity also increased 1.67- to 2.8-fold relative to the empty effector vector (Figure 4B). To confirm that StTCTP exhibits transcriptional activation activity, we constructed *pHIS-proStSN2* and *pGADT7-StTCTP* vectors and both the constructs were transformed into yeast cells. Positive clones grew on SD medium lacking Trp, His, and Leu and containing 3-AT (Figure 4E). Yeast one-hybrid assay further supported a role for StTCTP in activating *StSN2* gene expression. These data indicate that StTCTP functions as a transcriptional activator of *StSN2*.

### 2.4. StTCTP Was Upregulated by Drought Stress

TCTP is a protein that is highly conserved in evolution. Through systematic phylogenetic analysis of different plants, we found that the TCTP of potato is located on the same evolutionary branch as that of tomato, indicating their closest kinship. A further comparison of the amino acid sequences of these two proteins revealed that the amino acid sequence identity between potato TCTP and tomato TCTP is as high as 90.48% (Appendix A). It has been reported that overexpression of *SlTCTP* in tobacco can significantly increase the tolerance of transgenic seedlings to abiotic stress [31]. Considering that the *TCTP* of potato and tomato share high amino acid sequence consistency and close evolutionary relationships, it is not unreasonable to suggest that their biological functions may be similar. The initial verification of the response of *StTCTP* to drought treatment was conducted by examining the expression levels in WT leaves on days 0, 4, 8, and 12 following drought treatment. Similar to *StSN2*, the expression level of *StTCTP* rapidly increased under drought treatment. Transcription levels on day 12 were 10 times higher than on day 0 (Figure 5A). Spatial expression analysis showed that the expression levels of *StTCTP* in leaves on the 12th day after drought treatment were 10 times and 4 times higher than those in stem and root, respectively (Figure 5B). From this, we can deduce that *StTCTP* is also a gene that responds to drought stress.

To explore the roles of *StTCTP* in drought tolerance, we overexpressed *StTCTP* in the potato cultivar ‘Chuanyu 10’ and demonstrated that *StTCTP* was highly expressed in lines *OE-T1*, *OE-T7*, and *OE-T8* (Figure 5C). StTCTP protein accumulation was confirmed using Western blot (Figure 5D). Interestingly, we also found that *StTCTP* overexpression promoted the expression of *StSN2* compared with WT plants (Appendix A). This observation further supports the results from our dual luciferase and yeast one-hybrid assays.

### 2.5. Overexpression of StTCTP Enhances Tolerance to Drought

To explore the contribution of *StTCTP* to drought tolerance, we treated 3-week-old WT, *OE-T1*, and *OE-T8* plants with progressive drought stress for 5 days using 5% PEG. We evaluated the growth inhibition effect of potato seedlings grown for 5 days in peat soil containing 5% PEG-6000. The *OE-T1* and *OE-T8* lines developed longer stems and more severe leaf drooping, yellowing, and blackened tip development than WT plants. By the 5th day of PEG treatment, the top leaves of the WT plants exhibited severe wilting, whereas the transgenic plants were still growing quite well (Figure 6A). Drought stress can weaken plant photosynthesis, leading to a decrease in the plant’s photosynthetic rate [32]. Our research found that, as drought increases, the degree of wilting in plants intensifies; however, the growth of *OE-T1* and *OE-T8* is superior to that of WT (Figure 6A). The fluorescence results of the leaves were consistent with the degree of wilting observed in the plants (Figure 6B). We further quantified the drought stress response of the WT and overexpression lines by measuring the photosynthetic capacity of the plants. Generally speaking, the Fv/Fm value of normal plants ranges between 0.75 and 0.85. Chlorophyll a fluorescence (Fv/Fm) values of functional leaves at different times after stress induction revealed that the Fv/Fm values decreased for all leaves under stress. However, the Fv/Fmvalues in leaves overexpressing *StTCTP* remained higher than those of WT plants. On the fifth day of treatment, the Fv/Fm values of WT plant leaves were approximately 60% of those from the overexpression lines (Figure 6C). The results of water loss and RWC were similar to *StSN2*. RWC of *OE-StTCTP* was higher than WT, and water loss was lower (Figure 6D,E).

Excessive production and accumulation of ROS induced by stress can lead to increased MDA accumulation and can cause damage to proteins, DNA, and other cellular components [33]. To better assess the role of *StTCTP* in limiting cellular damage caused by oxidative stress, we quantified H_2_O_2_ and MDA accumulation and CAT enzyme activity. MDA and H_2_O_2_ accumulation were significantly lower in *OE-T1* and *OE-T8* plants compared with WT after 5 d of drought treatment (Figure 7A,B). CAT activity in *OE-T1* and *OE-T8* was higher than in WT during drought stress (Figure 7C). These results suggest that *StTCTP* overexpression reduces oxidative stress caused by drought in potatoes. In general, *StTCTP* plays an important role in responding to drought stress.

## 3. Discussion

Drought inhibits cell growth, leading to a significant annual reduction in crop yield. Between 2006 and 2015, the average annual economic loss in China due to drought was approximately USD 12.8 billion, which accounted for 0.16% of the country’s gross domestic product (GDP) [34]. Potatoes, the fourth-largest food crop globally, are highly affected by drought, but drought-resistant varieties are rare due to their narrow genetic background. Studying drought resistance mechanisms in potatoes is crucial for boosting yield and quality. Drought stress triggers ROS production in plants, disrupting the oxidant–antioxidant balance and causing oxidative stress [33,35]. Research shows that cysteine sites in Snakin/GASA proteins form disulfide bonds vital for redox regulation [8]. Overexpression of related proteins like *GIP2* in *petunia* and *FsGASA4* in *Arabidopsis* reduces oxidative stress and enhances tolerance to various stresses [9,36]. StSN2, an antibacterial peptide in potatoes belonging to the Snakin/GASA family [37], exhibits a specific expression pattern under drought stress, with its transcription level rising as stress intensifies (Figure 1B,C). Thus, *StSN2* is a key drought-responsive gene in potatoes.

In this study, we found that the overexpression of *StSN2* conferred better drought tolerance than the silencing lines, which was evident in plant height and root length (Figure 2A–C). H_2_O_2_ serves as a crucial reactive molecule in plants under drought stress [38]. MDA is the final product of lipid peroxidation reaction, and its level can be an important indicator for evaluating the degree of cell membrane damage. Under PEG-6000 treatment, *StSN2* suppresses the accumulation of H_2_O_2_ and MDA within cells, thereby enhancing the drought tolerance of potato (Figure 2B,D). To maintain intracellular ROS homeostasis, plants have evolved a sophisticated antioxidant system. In our investigations, we observed that, compared with WT plants, overexpression of *StSN2* led to increased activities of CAT, SOD, and POD under drought stress conditions (Figure 3C,E,F). Conversely, the *RNAi-StSN2* lines exhibited an opposite pattern. Additionally, DAB staining confirmed that *StSN2* indeed inhibits the accumulation of ROS (Figure 3A). These findings suggest that *StSN2* plays a significant role in regulating the antioxidant system and mitigating oxidative damage in potato under drought stress. Meanwhile, previous studies have also shown that overexpression of *StSN2* promotes SOD and CAT enzyme activity and inhibits the accumulation of H_2_O_2_ in buds [13]. In our previous research, we also observed a similar phenomenon, where *StSN2* suppressed the accumulation of H_2_O_2_ in the bud eyes [39]. The water loss and relative water content of detached leaves are usually negatively correlated. *StSN2* effectively enhanced potato tolerance to drought by reducing leaf water loss and increasing leaf relative water content (Figure 2D,E). Based on the above, it can be concluded that *StSN2* inhibits the accumulation of ROS by increasing the activity of antioxidant enzymes, thereby improving the drought resistance of potato.

The TCTP family members are highly conserved proteins that play a crucial role in plant growth, development, and stress response mechanisms. In plants, the transcription level of *TCTP* varies in response to salt stress, high temperature, and drought. Heat stress has been reported to induce the transcription of *TCTP* in Jatropha curcas and cabbage [22,40]. Additionally, in *Hevea brasiliensis*, the transcription level of *TCTP* rapidly increases after exposure to drought [24]. In our research, we discovered that the transcription level of *StTCTP* rose consistently as drought stress intensified (Figure 5A). *StTCTP* behaves similarly to *StSN2* and serves as a gene that responds to drought stress. CAT is the main enzyme involved in ROS detoxification. Overexpression of *StTCTP* promotes the activity of CAT under drought stress, leading to the suppression of H_2_O_2_ and MDA accumulation (Figure 7A–C). Drought stress has a significant inhibitory effect on photosynthesis in plants, leading to a significant decrease in the photosynthetic rate. The Fv/Fm value is a key indicator for assessing plant photosynthesis and stress resistance, reflecting the maximum photosynthetic capacity of plants [41]. Previous studies have found that overexpression of *SlTCTP* in tobacco increases the photosynthetic rate of transgenic plants, and the biomass of overexpressed plants also significantly increases [33]. In this study, we observed that the Fv/Fm values of *StTCTP* overexpressing plants were significantly higher than those of WT (Figure 6C). This suggests that *StTCTP* can mitigate the inhibition of photosynthesis by drought, consistent with previous research. In addition, overexpression of *StTCTP* can also reduce water loss in leaves and maintain relative water content, and improve drought tolerance (Figure 6D,E). Therefore, we propose that *StTCTP* plays a positive role under drought stress conditions.

*StSN2* and *StTCTP* play important roles in responding to drought stress. DBAS screening results indicate that StTCTP is an upstream regulator of *StSN2*. Yeast one-hybrid, electrophoretic mobility shift assays, and luciferase reporting assay show that the StTCTP protein can bind to the *StSN2* promoter. LUC enzyme activity detection results suggest that StTCTP can activate the *StSN2* promoter. In addition, we found that the expression level of *StSN2* was significantly higher in the overexpression lines of *StTCTP* than in WT. Based on the above results, we conclude that StTCTP binds to the *StSN2* promoter to promote the expression of *StSN2.*

In summary, we have demonstrated that the Snakin/GASA family gene *StSN2* is a drought-responsive gene. *StSN2* is specifically expressed in potatoes under drought treatment. Transgenic materials indicate that *StSN2* can delay the toxic effects of ROS on cells by increasing the activity of antioxidant enzymes under drought stress. In the screening of upstream regulatory elements of *StSN2*, StTCTP can target the promoter sequence of *StSN2* and activate its expression. Additionally, numerous clues suggest that *StTCTP* can eliminate the excessive accumulation of ROS and enhance photosynthesis in plants, thereby conferring high drought tolerance to plants. Given that both *StTCTP* and *StSN2* can increase drought tolerance, it is necessary to further investigate whether they act synergistically or independently. Furthermore, we believe that *StSN2* and *StTCTP* may even reduce ROS accumulation caused by other forms of abiotic stress, and we look forward to studying this in future research.

## 4. Material and Method

### 4.1. Plant Materials

The open reading frame (507 bp) of *StTCTP* (Soltu. DM. 01G039500.1) was inserted into the *pCAMBIA-2300-GFP* vector driven by the 35S promoter. The recombinant vectors were transformed into the GV3101 strain. Strains were cultured in YEB liquid medium and shaken at 28 °C overnight. The potato stem segments (approximately 0.5–1 cm length) of ‘Chuanyu 10’ (WT) were immersed in a bacterial suspension for 5–8 min and incubated in dark conditions for 36 h. Then, the stem segments were transferred to a shoot-differentiation medium for shooting induction [42].

Sterile seedlings were grown at 20 °C in a 16 h/8 h light and dark cycle. Dual luciferase assays were performed in *Nicotiana benthamiana* L. Tobacco seeds were planted in soil and raised in a growth chamber at a temperature of 20 °C in a 16 h/8 h light and dark cycle [43].

### 4.2. Drought Tolerance Experiment

We transferred WT potato tissue culture seedlings with a length of 6–8 cm into pots filled with coconut bran and cultivated them in a growth chamber at a temperature of 20 °C in a 16 h/8 h light and dark cycle for 60 days. Subsequently, the seedlings were exposed to natural drought stress for 12 days. Three biological replicates were performed.

### 4.3. PEG-Treated Seedlings Experiment

Polyethylene glycol (PEG) is an osmotic regulator that simulates drought stress in plants [44]. Shoot tips (2–3 cm) from *StSN2* transgenic and WT potato tissue-cultured seedlings were cut and transferred onto fresh MS plates supplemented with or without PEG-6000. Shoot tips were then placed in the growth chamber for 4 weeks.

*StTCTP* transgenic and WT tissue cultured seedlings were planted in flower pots with coconut bran. Pots were placed in the growth chamber for three weeks. After 3 weeks, seedlings were irrigated with 30 mL of 5% PEG-6000 each day for 5 consecutive days.

### 4.4. Water Loss and Relative Water Content Measurements

Water loss and relative water content (RWC) measurements were conducted using a previously described method [45]. First, 20 pieces of leaves from both the WT, OE-StSN2 lines, *RNAi-StSN2* lines, and *OE-StTCTP* lines were collected. The functional leaves (the third to fourth functional leaves from the top of the potato plants) of potatoes were detached and weighed. The weight was measured every 0.5 h over 2.5 h. The rate of water loss was calculated by the loss of fresh weight (FW). For the RWC assay, we measured the FW of the functional leaves. Next, the leaves were immersed for 12 h in distilled water to measure their turgid weight (TW), then dried at 65 °C to measure their dry weight (DW). RWC (%) = (FW − DW)/(TW − DW).

### 4.5. Measurement of Indices of Drought Stress Tolerance

Staining with DAB was performed using a previously described method [46]. First, the leaves were thoroughly cleaned, then immersed in DAB staining solution and subjected to overnight dark treatment. Next, the leaves were bleached using alcohol. Finally, photographs of the bleached leaves were taken for observation.

All samples were prepared for enzyme activity by homogenizing 0.1 g of leaf in a solution of 0.01 mM pH 7.2 phosphate buffer saline. The homogenate was centrifuged at 12,000 rpm for 10 min at 4 °C. The activities of SOD, CAT, and POD were measured separately by using a SOD assay kit (Cat. BC0175), CAT assay kit (Cat. BC0205), and POD assay kit (Cat. BC0095) produced by Solarbio life science. The levels of MDA and H_2_O_2_ were performed based on the procedure described in the manufacturer’s directions (Solarbio, Beijing, China) [47,48,49].

### 4.6. DNA Pull-Down

A total of 100 μg of a biotin-conjugated promoter fragment and avidin magnetic beads was placed in a 1.5 mL centrifuge tube and placed on a horizontal shaker set to 200 rpm for 3–4 h. Following incubation, 1 mL of potato budding nucleoprotein extract was added and incubated with the promoter and beads at 200 rpm of shaking at 4 °C for 6–8 h. Centrifugation at 3000× *g* for 5 min was performed, followed by three rounds of elution and precipitation in 1xPBS. Protein was separated using 10% SDS-PAGE and an unlabeled promoter fragment was added to the control. The silver staining method was used to detect proteins in the polyacrylamide gel [50]. Specific protein bands were excised and identified by LC-MS/MS analysis using Beijing Bio-Tech Pack Technology Company Ltd. (BTP) (Beijing, China). DNA pull-down was conducted following a previously described method [51].

### 4.7. Luciferase Reporting Assay

*StTCTP* was subcloned into the *pGreenII 62-SK* vector, and the *StSN2* promoter was linked in the *pGreenII 0800-LUC* vector. The *StTCTP* and *StSN2* promoter constructs were transformed into *Agrobacterium tumefaciens* strain GV3101. The *Agrobacterium suspensions* carrying the *StTCTP* and *StSN2* promoter constructs were prepared for tissue transformation by incubating in YEB and shaking at 28 °C and 200 rpm overnight. The next day, the Agrobacterium suspensions carrying the indicated constructs were co-infiltrated into tobacco leaves and placed back in the growth chamber. After 36 h, luminescence was photographed using living plant imaging systems (Viber Fusion FX, Paris, France), and the dual luciferase reporter assay kit (Vazyme, Nanjing, China) was used to measure LUC enzyme activity [52]. The primers used are listed in Appendix A. This experiment was performed with three independent biological replicates.

### 4.8. Yeast One-Hybrid Assays

The full length sequence of the *StTCTP* gene was ligated into the *pGADT7* vector. The promoter of *StSN2* was subcloned into the *pHIS* reporter vector to create *pHIS-proStSN2*, which was transformed into Y187 yeast cells. Yeasts were plated on SD/-Leu/-Trp with 3-aminotriazole (0 mM, 30 mM, 60 mM, 90 mM) media to determine the optimal concentration of 3-aminotriazole for screening. The *pHIS-proStSN2* and *pGADT7-Rec2-StTCTP* were transformed into Y187 yeast cells. The empty vector *pGADT7* and *pHIS-proStSN2* were co-transformed into Y187 yeast as negative controls. Yeast was plated on (SD/-Leu/-Trp/-His) media supplemented with 3-aminotriazole (0 mM, 30 mM, 60 mM, 90 mM) to select the positive clones.

### 4.9. Electrophoretic Mobility Shift Assays

The *StTCTP* gene was subcloned into the *pCold-TF* vector to produce a StTCTP-His expression vector driven by a *cspA* promoter, and the recombinant plasmid was transformed into BL21 *Escherichia coli*. Expression of the His-tagged and His-StTCTP fusion proteins was induced by 1 mM isopropyl-b-D-thiogalactoside (IPTG) at 16 °C for 20 h, and then the fusion proteins were purified with Ni-NTA agarose. The 5′FAM-labeled oligonucleotide probes were directly synthesized and labeled by the BeijingTsingke company. EMSAs were carried out according to the protocol provided with a chemiluminescent EMSA kit (GS009, Beyotime Biotechnology, Shanghai, China). Finally, the FAM-labeled DNA on the gel was detected on a ChemiDoc XRS system (Bio-Rad, Hercules, CA, USA).

### 4.10. Bioinformatics Analysis

The potato database (https://spuddb.uga.edu/ (accessed on 18 March 2024)) was used to obtain the full-length gene of *StTCTP* and the *StSN2* promoter. Snakin/GASA and TCTP homologous proteins from A. thaliana and other plants were downloaded from the NCBI database (https://www.ncbi.nlm.nih.gov/ (accessed on 20 March 2024)). DNAMAN 6.0 was used to evaluate the homology of amino acid sequences. Phylogenetic analysis was performed using the neighbor-joining method in MEGA7.0 software with 2000 replications.

### 4.11. Expression Analysis

We used the MolPure^®^ Plant RNA Kit (Yeasen, Shanghai, China, Cat. 19291ES50) to extract total RNA from potato roots, stems, and leaves. Hieff^®^ qPCR SYBR Green Master Mix (Yeasen, China, Cat. 11201ES08) was used for qRT-PCR. The qPCR assay was performed using a 7500 Real-Time PCR system (Bio-Rad, Hercules, CA, USA). The relative expression level of genes was calculated using the 2^−ΔΔCt^ method and elongation factor 1 α-like (EF-1α) as a reference gene [53]. The experiments were conducted using three biological replicates. The primer sequences used can be found in Appendix A.

### 4.12. Western Blotting

Anti-GFP (Cat.GB15603-100) and anti-actin (Cat. GB15001-100) antibodies were purchased from Servicebio. A total of 20 μg of protein from the tuber budding eyes was collected for each sample. Proteins were separated using 10% SDS-PAGE followed by a transfer of proteins to a nitrocellulose membrane using the wet transfer method [54]. The membrane was immersed in a 5% skim milk buffer for 2 h. The primary anti-GFP and anti-actin antibodies were added at a 1:1000 ratio and incubated at 4 °C overnight. The next day, a secondary antibody was added at a ratio of 1:1000 (Servicebio, Wuhan, China, Cat. GB23204) for 2 h. Chemiluminescence was detected using a BeyoECL Plus Kit (Beyotime, Shanghai, China) [55].

### 4.13. Measurement of Chlorophyll a Fluorescence

Before the measurement, the potato seedlings needed to be placed in dark conditions for 30 min. Chlorophyll a fluorescence was recorded following 1 s of light exposure. The potential photosynthetic efficiency (FV/FM) was recorded by taking the variable fluorescence (FV) and dividing it by the maximal fluorescence (FM) [56]. Chlorophyll a fluorescence was measured using an IMAGING-PAM-MAXI chlorophyll fluorescence imaging system (Heinz Walz GmbH, Effeltrich, Germany) [57].

### 4.14. Statistical Analysis

Statistical analysis and figure plotting were conducted using SPSS 24.0 and Origin 2021 software. All experiments in this study underwent three biological replicates and the data are shown as mean ± SD (*n* = 3). Different letters in the figures indicate significant differences.

## Figures and Tables

**Figure 1 ijms-26-02796-f001:**
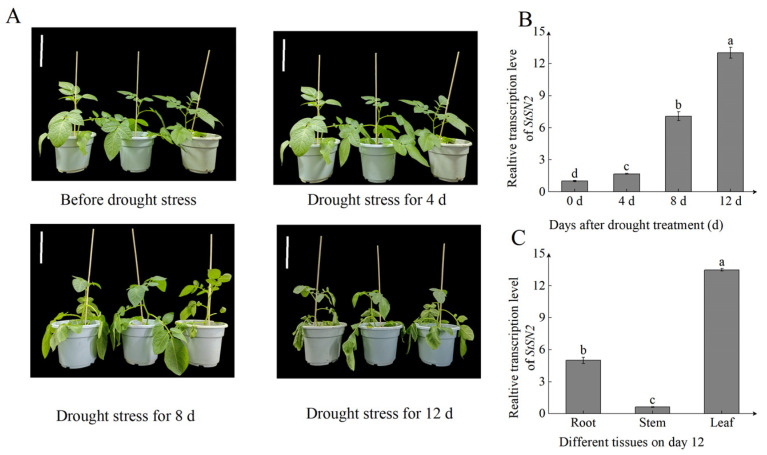
The effects of drought stress on WT potatoes and the expression levels of the *StSN2* gene at the same time. (**A**) Growth status of potatoes under drought treatment. Scale bar = 10 cm. (**B**) The expression level of the *StSN2* gene under drought treatment. (**C**) The spatial expression of potato *StSN2* on the 12th day of drought treatment. Data are means ± SD of three biological replicates. Different lowercase letters indicate significant differences for the same treatment (*p* < 0.05).

**Figure 2 ijms-26-02796-f002:**
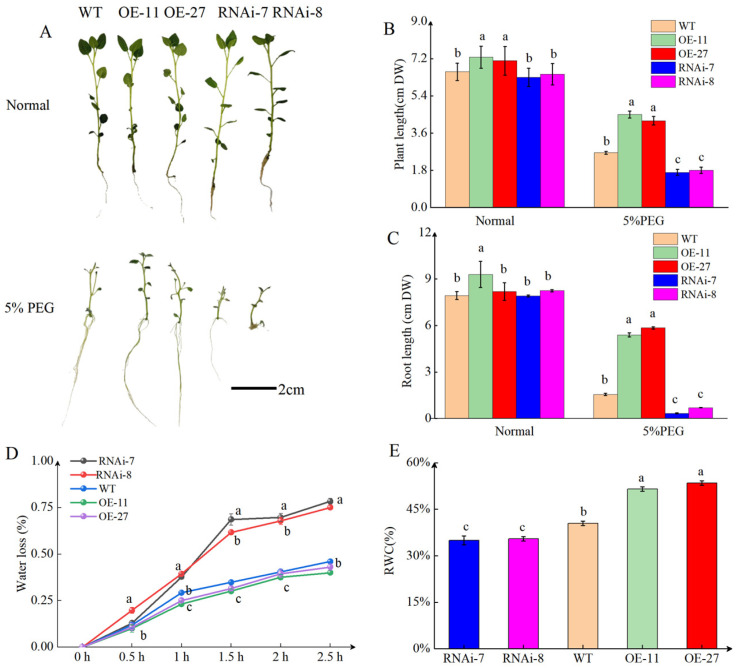
Overexpression of *StSN2* enhances tolerance to drought stress in potato. (**A**) Images of transgenic *OE-StSN2*, *RNAi-StSN2*, and WT potato lines after four weeks of growth on MS medium in normal conditions or 5% PEG-6000. Scale bar = 2 cm. (**B**) Quantification of plant length. (**C**) Quantification of root length. (**D**) Rates of water loss in detached leaves of seedlings from *OE-StSN2*, *RNAi-StSN2*, and WT were measured every 0.5 h over a total of 2.5 h. (**E**) Leaf RWC of *OE-StSN2*, *RNAi-StSN2*, and WT potato plants. Data are means ± SD of three biological replicates. Different lowercase letters indicate significant differences for the same treatment (*p* ≤ 0.05).

**Figure 3 ijms-26-02796-f003:**
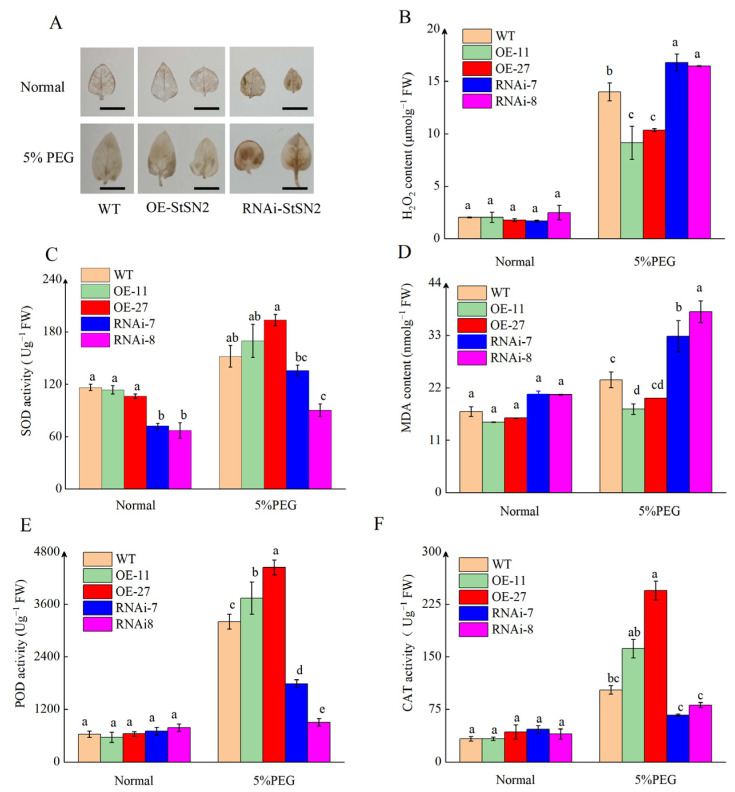
Overexpression of *StSN2* in potato reduces ROS accumulation during drought. (**A**) *OE-StSN2*, *RNAi-StSN2*, and WT potato lines were grown on MS medium in normal conditions or 5% PEG-6000 for four weeks. A DAB staining. Scale bar = 1 cm. (**B**) Quantification of H_2_O_2_ accumulation. (**C**) SOD activity. (**D**) MDA accumulation. (**E**) POD activity. (**F**) CAT activity. Data are means ± SD of three biological replicates. Different lowercase letters indicate significant differences for the same treatment (*p* ≤ 0.05).

**Figure 4 ijms-26-02796-f004:**
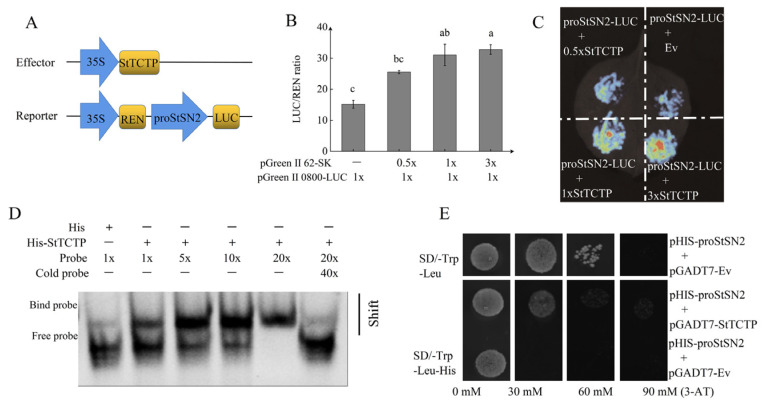
StTCTP and *StSN2* promoter interaction verification. (**A**–**C**) Dual luciferase assays. (**D**) EMSA assay. (**E**) Yeast one-hybrid assay. Data are means ± SD of three biological replicates. Different lowercase letters indicate significant differences for the same treatment (*p* ≤ 0.05).

**Figure 5 ijms-26-02796-f005:**
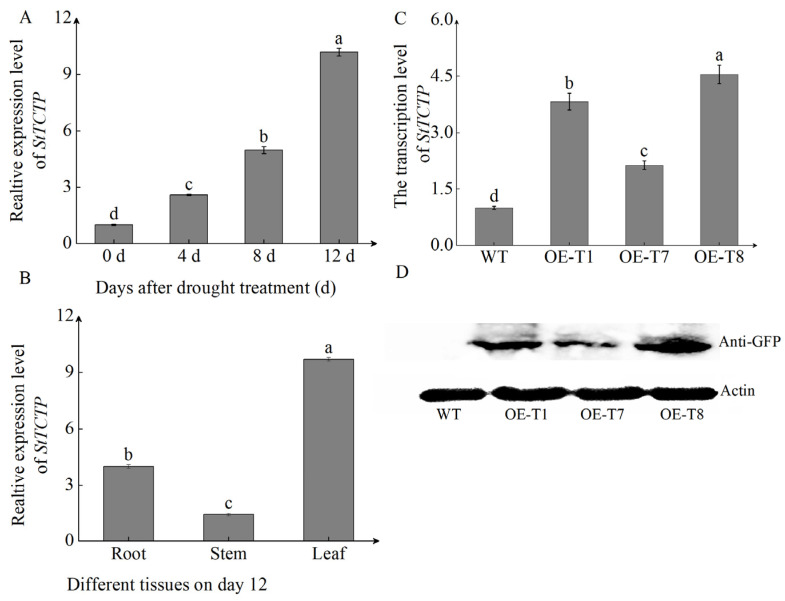
The expression levels of the *StTCTP* gene in potato. (**A**) Expression of *StTCTP* after drought treatment. (**B**) Tissue expression of *StTCTP* on the 12th day of drought treatment. (**C**) *StTCTP* expression in WT and *OE-StTCTP* lines plants. (**D**) StTCTP protein accumulation in WT and *OE-StTCTP* lines. Actin was used as an internal protein loading control. Data are means ± SD of three biological replicates. Different lowercase letters indicate significant differences for the same treatment (*p* ≤ 0.05).

**Figure 6 ijms-26-02796-f006:**
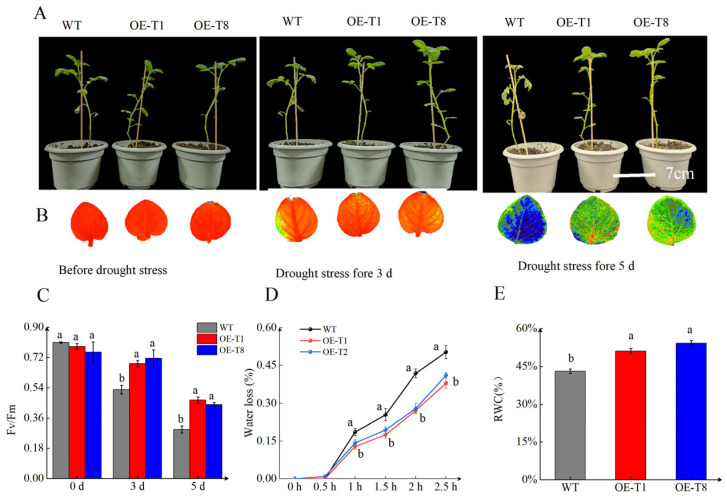
Overexpression of *StTCTP* enhances tolerance to drought stress. (**A**) Image of *OE-StTCTP* and WT potato lines growing in 5% PEG-6000. Scale bar = 7 cm. (**B**) Fluorescence imaging of chlorophyll in potato leaves under drought stress. (**C**) Chlorophyll a fluorescence values (Fv/Fm). (**D**) Rates of water loss in detached leaves of seedlings from WT and *OE-StTCTP* potato plants were measured every 0.5 h over a total of 2.5 h. (**E**) Leaf RWC of WT and *OE-StTCTP* potato plants. Data are means ± SD of three biological replicates. Different lowercase letters indicate significant differences for the same treatment (*p* ≤ 0.05).

**Figure 7 ijms-26-02796-f007:**
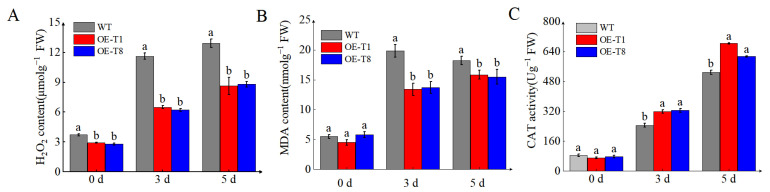
ROS accumulation and CAT activity in leaves of potatoes. (**A**) H_2_O_2_ content. (**B**) MDA content. (**C**) CAT enzyme activity. *OE-StTCTP* and WT potato lines were treated for 5 days with 5% PEG-6000. Data are means ± SD of three biological replicates. Different lowercase letters indicate significant differences for the same treatment (*p* ≤ 0.05).

## Data Availability

The data presented in this study are available within the article.

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
