# Peer review of "StTCTP Positively Regulates StSN2 to Enhance Drought Stress Tolerance in Potato by Scavenging Reactive Oxygen Species"

_ijms, 2025, doi:10.3390/ijms26062796_

Round 1
Reviewer 1 Report
Comments and Suggestions for Authors
The authors investigated the mechanism by which StTCTP enhances the drought stress tolerance of potatoes through the positive regulation of StSN2, which provides a theoretical basis for breeding drought-tolerant potato varieties. Overall, the research is innovative and valuable, but there are still some details that need to be improved. The comments are as follows:
The writing of the manuscript needs to be further improved. Currently, there are issues such as unclear expressions and grammatical errors. Please have professionals with excellent English skills revise the article.
The Latin name and gene name need to be in italics.
Please double-check the significance analysis. There are instances where the data appear to have a large disparity, yet the difference is not statistically significant.
Please specify which generation of transgenic plants was utilized.
Please supplement the transcription level and protein abundance of target gene in the leaves of overexpression lines.
In the abstract, abbreviations need to be spelled out in full. For example, SN2 and RWC.
Introduction
There should be a full stop at the end of the first sentence.
Paragraph 2: Please introduce the relationship between SN2 and Snakin/GASA.
Line 58: AtGASA4?
Last paragraph: The background knowledge has already been introduced earlier in the paper. I recommend deleting it to streamline the content and avoid repetition.
Lines 105-111: Please briefly illustrate the significance of this research.
Results
Line 117: “More than 50days” is too broad a range. Please be more precise.
Line 122: As shown in Figure 1B, the expression level changed significantly on the fourth day.
Line 149, line 151: It seems that authors cited the wrong figure. Please ensure that the order in which the figures appear is consistent with the order described in the main text.
Figure 1D and Figure 6D: Please add the significance analysis.
Figure 1D, E: Were the data acquired under normal or drought conditions? Please clarify.
There is an extra full stop in Line 187.
Line 188: Delete “they”.
Lines 194-198: Please cite the correct figure. And the results of Y1H were mentioned repeatedly.
Figure 4: The legends for sub-panels D and E are missing. What does 0.5StTCTP mean? It seems that StTCTP was over-expressed by the 35S promoter.
Figure 4D: This experiment needs improvement. A shifted band can be seen when only His and Probe are added, suggesting non-specificity of the probe. Additionally, when 20 Probes are added, the free probes disappear.
Figure 5D: Please show the marker.
Figure 7A: Please show the DAB staining image. It is recommended to supplement the enzyme activities of SOD and POD.
Discussion
The background knowledge in the first two paragraphs is repetitive with that in the introduction. Please remove or simplify it.
Lines 317-318: Instead of “In my previous research, I also observed…”, it is better to use “In our previous research, we also observed…”.
Line 363: According to the title of this MS, StTCTP regulates StSN2.
4.11 and 4.12 are the same.
Supplementary Fig. S4: Change “full length” to “full-length”
Supplementary Table 1: The bottom line of this table is missing.
Supplementary Fig. S1: Please use the data from leaves.
Comments on the Quality of English Language
The overall expression can be further refined.
Author Response
Dear Ms. teacher After receiving your final response, I offer my sincerest apologies for the errors present in the manuscript. I have carefully reviewed your feedback and am in awe of your meticulous scientific approach. Moreover, I am grateful for the opportunity you have given me to resubmit my manuscript. Now we are submitting the revised manuscript entitled “StTCTP positively regulates StSN2 to enhance drought stress tolerance in potato by scavenging reactive oxygen species” (ID: ijms-3515110) for consideration for publication in《International Journal of Molecular Sciences》. In the revision process, we have implemented nearly all the suggestions and addressed all the comments made by the reviewers. We appreciate the time and effort that you and the reviewers dedicated to providing feedback on our manuscript. We are grateful for the insightful comments and valuable improvements to our paper. Those comments are valuable for revising and improving our paper with important guiding significance.We have made corrections according to the comments. The revised portions are marked in yellow on the paper. All page numbers refer to the revised manuscript file with tracked changes. I hope that the changes I’ve made resolve all your concerns about the article. I’m more than happy to make any further changes that will improve the paper and/or facilitate successful publication. Reviewer 1 The writing of the manuscript needs to be further improved. Currently, there are issues such as unclear expressions and grammatical errors. Please have professionals with excellent English skills revise the article. The Latin name and gene name need to be in italics. Please double-check the significance analysis. There are instances where the data appear to have a large disparity, yet the difference is not statistically significant. Please specify which generation of transgenic plants was utilized. Please supplement the transcription level and protein abundance of target gene in the leaves of overexpression lines. In the abstract, abbreviations need to be spelled out in full. For example, SN2 and RWC. Introduction There should be a full stop at the end of the first sentence. Paragraph 2: Please introduce the relationship between SN2 and Snakin/GASA. Line 58:AtGASA4? Last paragraph: The background knowledge has already been introduced earlier in the paper. I recommend deleting it to streamline the content and avoid repetition. Lines 105-111: Please briefly illustrate the significance of this research. Results Line 117: “More than 50days” is too broad a range. Please be more precise. Line 122: As shown in Figure 1B, the expression level changed significantly on the fourth day. Line 149, line 151: It seems that authors cited the wrong figure. Please ensure that the order in which the figures appear is consistent with the order described in the main text. Figure 1D and Figure 6D: Please add the significance analysis. Figure 1D, E: Were the data acquired under normal or drought conditions? Please clarify. There is an extra full stop in Line 187. Line 188: Delete “they”. Lines 194-198: Please cite the correct figure. And the results of Y1H were mentioned repeatedly. Figure 4: The legends for sub-panels D and E are missing. What does 0.5StTCTP mean? It seems that StTCTP was over-expressed by the 35S promoter. Figure 4D: This experiment needs improvement. A shifted band can be seen when only His and Probe are added, suggesting non-specificity of the probe. Additionally, when 20 Probes are added, the free probes disappear. Figure 5D: Please show the marker. Figure 7A: Please show the DAB staining image. It is recommended to supplement the enzyme activities of SOD and POD. Discussion The background knowledge in the first two paragraphs is repetitive with that in the introduction. Please remove or simplify it. Lines 317-318: Instead of “In my previous research, I also observed…”, it is better to use “In our previous research, we also observed…”. Line 363: According to the title of this MS, StTCTP regulates StSN2. 4.11 and 4.12 are the same. Supplementary Fig. S4: Change “full length” to “full-length” Supplementary Table 1: The bottom line of this table is missing. Supplementary Fig. S1: Please use the data from leaves. Comment 1: The writing of the manuscript needs to be further improved. Currently, there are issues such as unclear expressions and grammatical errors. Please have professionals with excellent English skills revise the article. Response 1: We apologize for our poor writings. Thank you for your suggestion! we once again invited a friend of us who is a native English speaker from the USA to help proofread this manuscript. We have carefully and thoroughly proofread the manuscript to correct all the grammar and spelling mistakes. And we hope the revised manuscript could be acceptable for you. Comment 2:The Latin name and gene name need to be in italics. Response 2 :We apologize for the issues in the manuscript. Based on the reviewer's suggestion, we thoroughly reviewed the manuscript and italicized the genes and Latin names in the manuscript, highlighted them in yellow. There are too many citations involved, so I will not list them here. Thank you. Comment 3: Please double-check the significance analysis. There are instances where the data appear to have a large disparity, yet the difference is not statistically significant. Response 3 : We are immensely grateful to the reviewers for bringing this issue to our attention. The statistical analysis throughout our entire manuscript was conducted using the DPS software. Upon re-analyzing the previous data, we found errors in the significance labeling of plant heights for the different materials under 5% PEG treatment in Figure 2-B and have revised the labeling accordingly. Similarly, we identified errors in the significance labeling of root lengths for the different materials under normal growth conditions in Figure 2-C and have also corrected those labels. No abnormalities have been detected in the other figures at this time. We sincerely thank the teachers for their diligent work. As follows: Figure 2. Overexpression StSN2 enhances tolerance to drought stress in potato. (A) Images of transgenic OE-StSN2, RNAi-StSN2, and WT potato lines after four weeks of growth on MS medium in normal conditions or 5% PEG-6000. Scale bar=2 cm. (B) Quantification of plant length. (C) Quantification of root length. (D) Rates of water loss in detached leaves of seedlings from OE-StSN2, RNAi-StSN2, and WT were measured every 0.5 h over a total of 2.5 h. (E) Leaf RWC of OE-StSN2, RNAi-StSN2, and WT potato plants. Data are means±SD of three biological replicates. Different lowercase letters indicate significant differences for the same treatment (P≤0.05). Comment 4: Please specify which generation of transgenic plants was utilized. Response 4 : I am extremely grateful to the reviewer for raising this question. Potatoes reproduce asexually, and in our experiments, we infected potato stem segments with Agrobacterium, and after verifying their correctness, proceeded with subsequent experiments. Therefore, it is not possible to specify which generation our experiments were conducted on. Prior to the experiments, we conducted identifications to ensure that no genes were lost. I hope my explanation addresses your concerns. Thank you very much! Comment 5: Please supplement the transcription level and protein abundance of target gene in the leaves of overexpression lines. Response 5 :The StSN2 overexpression and silencing materials we used are transgenic materials successfully created in our laboratory. The gene expression levels and protein expression levels of these materials have been characterized in the article titled 《The Cysteine-Rich Peptide Snakin-2 Negatively Regulates Tubers Sprouting through Modulating Lignin Biosynthesis and H2O2 Accumulation in Potato》DOI:10.3390/ijms22052287 (Figure 2). In this experiment, I only conducted identification of the gene expression level of StSN2, with the relevant results presented in Supplementary Figure S1. Since the protein level of StSN2 had been previously characterized, I did not conduct another identification. The identification results for StTCTP are shown in Figure 5 of the manuscript. I hope my explanation addresses your concerns. Thank you! As follows: 《The Cysteine-Rich Peptide Snakin-2 Negatively Regulates Tubers Sprouting through Modulating Lignin Biosynthesis and H2O2 Accumulation in Potato》 (Figure 2) Figure 2. Comparison of gene and protein expression levels in sprouts of wild-type (WT) with transgenic plants either down-regulating (RNAi) or overexpressing (OE) the StSN2 gene. (A) Gene expression levels in transgenic lines. Data are means ± SD of three biological replicates. Different letters indicate significant differences at p < 0.05. (B) Protein expression levels measured by western blot (WB). The wild-type (WT), potato cultivar Chuanyu 10, was used as the control. Comment 6: In the abstract, abbreviations need to be spelled out in full. For example, SN2 and RWC. Response 6 : Thank you for this suggestion. I have made arrangements for the suggested rewrites of the abstract section As follows: (page1, line13-35) Abstract: Drought is a negative agronomic effect that can lead to an increase in reactive oxygen species (ROS) levels. Excessive can severely alter cell membrane fluidity and permeability, significantly reducing cell viability. Gibberellic acid-stimulated Arabidopsis (Snakin/GASA) gene family plays an important role as antioxidants in inhibiting the accumulation of ROS and improving crop drought resistance. However, the regulatory mechanism of potato StSnakin-2 (StSN2) in response to drought, as well as how StSN2 expression is regulated, is not well understood. In this study, we found that StSN2 was induced by drought. Overexpression of StSN2 significantly increased drought tolerance, whereas silencing StSN2 increased sensitivity to drought. Overexpression of StSN2 resulted in higher antioxidant enzyme (superoxide dismutase (SOD), catalase (CAT), and peroxidase (POD)) activity, and lowered hydrogen peroxide (H2O2) and malondialdehyde (MDA) accumulation during drought stress. Also, overexpression of StSN2 increased relative water content of leaves (RWC) and reduces water loss in leaves. We screened the upstream regulatory protein translation controlled tumour protein (StTCTP) of StSN2 through DNA pull-down combined with mass spectrometry. Yeast one-hybrid (YIH), electrophoretic mobility shift assays (EMSA) and luciferase reporting assay (LUC) indicated that StTCTP binds the StSN2 promoter. Like StSN2, StTCTP was highly expressed in response to drought. Overexpression of StTCTP increased the photosynthetic rate and CAT enzyme activity, and lowered H2O2 and MDA accumulation during drought. Meanwhile, overexpression of StTCTP increased leaf RWC and reduced water loss. Our research strongly suggested that StSN2 effectively cleared ROS and significantly boosted the drought resistance of potatoes. Furthermore, as a transcriptional activator of StSN2, StTCTP, much like StSN2, also enhanced the potato's drought tolerance. The result provided a foundation for further study of StSN2 regulatory mechanisms under drought stress. Comment 7: ①There should be a full stop at the end of the first sentence. ②Paragraph 2: Please introduce the relationship between SN2 and Snakin/GASA. ③Line 58: AtGASA4? ④Last paragraph: The background knowledge has already been introduced earlier in the paper. I recommend deleting it to streamline the content and avoid repetition. ⑤Lines 105-111: Please briefly illustrate the significance of this research. Response7 : Thank you for the suggestions from the teachers. ① I have added a period at the end of the first sentence in the Introduction (page 1, lines 39). ②In the second paragraph of the Introduction, I have clarified the relationship between StSN2 and Snakin/GASA (page 2, lines 66-67). StSN2 belongs to the Snakin/GASA family of proteins. ③Regarding AtGASA4, I had initially followed the naming convention from the literature title but have now corrected it in the manuscript (page 2, line 63-64). Overexpression of AtGASA4 and AtGASA14 in Arabidopsis resulted in reduced accumulation of ROS and enhanced stress tolerance ④The Introduction often contains summarizing statements in IJMS journal articles, and I have rewritten that paragraph accordingly, hoping to meet your expectations (page 2-3, lines 98-109). As follows: Potatoes, a key crop with high nutritional and economic value, often face drought stress during production[27]. The Snakin/GASA gene family, involved in plant stress responses[28], includes the drought-responsive gene StSN2, which is highly expressed in potato leaves under drought. StSN2 reduces ROS accumulation by boosting antioxidant enzyme activity, protecting cells from oxidative damage. Additionally, StTCTP was identified as interacting with the StSN2 promoter through DNA pull-down and mass spectrometry, confirmed by Y1H, EMSA, and LUC assays. Both StSN2 and StTCTP are drought-inducible, revealing a new drought response pathway in potatoes. These findings uncover a novel drought response mechanism in potatoes, offering key insights for developing drought-resistant varieties. Further study of StSN2 and StTCTP genes will clarify their roles in drought resistance, aiding the breeding of resilient potato varieties and providing valuable insights for drought research in other crops. Comment 8:Line 117: “More than 50days” is too broad a range. Please be more precise. Response 8 : We used to plant potato tissue culture seedlings in pots during natural drought stress treatment, and then processed them at 60 days. I did not express this clearly in the manuscript, please forgive me. I have made corrections in the manuscript (page 3, lines 114-115). We used the slow drought method to treat potato seedlings that had grown for 60 days. Comment 9:Line 122: As shown in Figure 1B, the expression level changed significantly on the fourth day. Response 9 : We apologize for the mistake we described. I have rewritten the paragraph again (page 3, lines 119-122). On the 0th and the 4th days, the expression of StSN2 increased rapidly. However, on the 8th and 12th days, the expression of StSN2 increased even more significantly, reaching levels 6.5 times and 12.8 times higher than on day 0, respectively (Figure 1B). Comment 10 :Figure 1D and Figure 6D: Please add the significance analysis. Response 10 :Thank you for the teacher's suggestion. I have reviewed the manuscript and found that there is no saliency annotation for images 2D and 6D, and there is no D-image in image 1. I conducted significance analysis on the 2D and 6D images according to the suggestions of the teachers and annotated them in the images. Because some points in the line chart are clustered together, I labeled them together. Figure 2. Overexpression StSN2 enhances tolerance to drought stress in potato. (A) Images of transgenic OE-StSN2, RNAi-StSN2, and WT potato lines after four weeks of growth on MS medium in normal conditions or 5% PEG-6000. Scale bar=2 cm. (B) Quantification of plant length. (C) Quantification of root length. (D) Rates of water loss in detached leaves of seedlings from OE-StSN2, RNAi-StSN2, and WT were measured every 0.5 h over a total of 2.5 h. (E) Leaf RWC of OE-StSN2, RNAi-StSN2, and WT potato plants. Data are means±SD of three biological replicates. Different lowercase letters indicate significant differences for the same treatment (P≤0.05). Figure 6. Overexpression of StTCTP enhances tolerance to drought stress. (A) Image of OE-StTCTP and WT potato lines growing in 5% PEG-6000. Scale bar=7 cm. (B) Fluorescence imaging of chlorophyll in potato leaves under drought stress. (C) Chlorophyll fluorescence values (Fv/Fm). (D) Rates of water loss in detached leaves of seedlings from WT and OE-StTCTP potato plants were measured every 0.5 h over a total of 2.5 h. (E) Leaf RWC of WT and OE-StTCTP potato plants. Data are means ±SD of three biological replicates. Different lowercase letters indicate significant differences for the same treatment (P≤0.05). Comment 11 : ①Figure 1D, E: Were the data acquired under normal or drought conditions? Please clarify. ②There is an extra full stop in Line 187. Response 11 :Thank you for the teacher's suggestion. ①There are no D and E in figure 1, it should be figure 2. The measurement of water loss and RWC indicators was carried out after PEG-6000 simulated drought stress treatment, which was also explained in the manuscript. (Pag4, lines 146-147) Figures 6D and 6E, The determination of water loss and RWC indicators was conducted after PEG-6000 simulated drought stress treatment. ②Thank you for the teacher's suggestion. I have already deleted this period (page5, line 187) Comment 12 : ①Line 188: Delete “they”. ②Lines 194-198: Please cite the correct figure. And the results of Y1H were mentioned repeatedly. ③Figure 4: The legends for sub-panels D and E are missing. What does 0.5StTCTP mean? It seems that StTCTP was over-expressed by the 35S promoter. Response 12 : Thank you for the teacher's suggestion. ①I have removed the word 'they'(page 5, line 188). ②-③ I have rewritten this paragraph (page 5-6, line 189-201). StTCTP was identified as a potential StSN2 promoter-binding protein (Supplementary Table 2). We validated this interaction using a dual luciferase assay in which the StSN2 promoter was placed upstream of luciferase (Figure 4A) and coexpressed with StTCTP in N.benthamiana leaves. Gradually increase in luminescence was observed when the StSN2pro-LUC was cotransformed with 35Spro-StTCTP relative to the empty effector vector (Figure 4C). LUC activity also increased 1.67 to 2.8-fold relative to the empty effector vector (Figure 4B). To confirm StTCTP exhibits transcriptional activation activity, we constructed pHIS-proStSN2 and pGADT7-StTCTP vectors, both the constructs were transformed into yeast cells. Positive clones grew on SD medium lacking Trp, His, Leu and containing 3-AT (Figure 4E). Yeast one-hybrid assay further supported a role for StTCTP in activating StSN2 gene expression. These data indicate that StTCTP functions as a transcriptional activator of StSN2. Figure 4. StTCTP and StSN2 promoter interaction verification. (A-C) Dual-luciferase assays (B) EMSA assay. (E) Yeast one-hybrid assay. Data are means±SD of three biological replicates. Different lowercase letters indicate significant differences for the same treatment (P≤0.05). Comment 13 : ①Figure 4D: This experiment needs improvement. A shifted band can be seen when only His and Probe are added, suggesting non-specificity of the probe. Additionally, when 20 Probes are added, the free probes disappear. ②Figure 5D: Please show the marker. ③Figure 7A: Please show the DAB staining image. It is recommended to supplement the enzyme activities of SOD and POD. Response 13: Thank you for the teacher's suggestion. I have made the following modifications: ① I strongly support the teacher's suggestion. We have conducted this EMSA experiment many times, and for your confusion, there are black objects (indicated by red arrows) in lanes 1 and 6, which are not bands. We have conducted this experiment many times, and both the protein and DNA have been purified. If we adjust the color difference, the thing pointed by the red arrow will disappear, but we don't want to do it that way as it appears unreal. As shown in the figure. I don't know if my explanation can dispel the teachers' concerns. ②The original image of 5D is as follows: ③Figure 7 shows the StTCTP drought stress treatment. We did not perform DAB staining on its leaves, but instead used a more intuitive chlorophyll fluorescence imaging system for photo processing (Figure 6B). In addition, we also detected the content of H2O2, which is more convincing than DAB staining. We measured the enzyme activity of CAT by measuring the content of H2O2, and the amount of MDA can reflect the degree of stress damage suffered by plants, which is a commonly used indicator in the study of plant aging physiology and resistance physiology. SOD, POD, and CAT are the three main enzymes in the plant antioxidant system. SOD and POD can indeed reflect the degree of drought stress in plants, but we measured the content of CAT and H2O2, which can basically reflect the plant's ability to resist stress. The teacher's suggestion is very correct. In future research, I will add these two indicators to better study the drought resistance of StTCTP. Comment 14 : ①The background knowledge in the first two paragraphs is repetitive with that in the introduction. Please remove or simplify it. ②Lines 317-318: Instead of “In my previous research, I also observed…”, it is better to use “In our previous research, we also observed…”. ③Line 363: According to the title of this MS, StTCTP regulates StSN2. ④4.11 and 4.12 are the same. ⑤Supplementary Fig. S4: Change “full length” to “full-length” ⑥Supplementary Table 1: The bottom line of this table is missing. 7Supplementary Fig. S1: Please use the data from leaves. Response 14: Thank you for the teacher's suggestion. I have made the following modifications: ①I have rewritten this paragraph as follows(page 8-9, line 284-298): Drought inhibits cell growth, leading to a significant annual reduction in crop yield. Between 2006 and 2015, the average annual economic loss in China due to drought was approximately 12.8 billion US dollars (USD), which accounted for 0.16% of the country’s gross domestic product (GDP)[34]. Potatoes, the fourth-largest food crop globally, are highly affected by drought, but drought-resistant varieties are rare due to their narrow genetic background. Studying drought resistance mechanisms in potatoes is crucial for boosting yield and quality. Drought stress triggers ROS production in plants, disrupting the oxidant-antioxidant balance and causing oxidative stress[33, 35]. Research shows that cysteine sites in Snakin/GASA proteins form disulfide bonds vital for redox regulation[8]. Overexpression of related proteins like GIP2 in petunia and FsGASA4 in Arabidopsis reduces oxidative stress and enhances tolerance to various stresses[9, 36]. StSN2, an antibacterial peptide in potatoes belonging to the Snakin/GASA family[37], exhibits a specific expression pattern under drought stress, with its transcription level rising as stress intensifies (Figure 1B-C). Thus, StSN2 is a key drought-responsive gene in potatoes. ② I have rewritten the sentence: In our previous research, we also observed a similar phenomenon where StSN2 suppressed the accumulation of H2O2 in the bud eyes[43] (page 9, line 315-316). ③ Teacher, this is correct. StTCTP can regulate the expression of StSN2. ④Thank you for pointing out, I have already deleted 4.12 ⑤ I have rewritten the sentence: Supplementary Fig. S4 Phylogenetic analysis and amino acid sequence alignment of StTCTP. A Phylogenetic tree analysis of Arabidopsis, tobacco, rubber, rice, tomato, corn, and other TCTP and StTCTP. StTCTP is indicated with a red dot. The full-length amino acid sequences were downloaded from NCBI database. B Comparison of amino acids in different species of TCTP. StTCTP is indicated with a red dot. ⑥ I have made the following modifications: Supplementary Table 1. Quantitative RT-PCR for RNA sequencing validation. Gene Purpose Forward/Reverse StSN2 qPCR TAACAGATGTAGCCACTGAC ACAACAAGTTCCACATGCCC Pro-StSN2 Yeast one-kybrid CGACTCACTATAGGGCGAATTCTGTAGTTGAACTTTTTATCA GACCGCGGATCGATTCGCGAACGCGTTGGAATTTGAAATATTTTTCT StSN2 Luciferase complementary CTATAGGGCGAATTGGGTACCTGTAGTTGAACTTTTTATCA TAGAACTAGTGGATCCCCCGGGTGGAATTTGAAATATTTTTC StSN2 Kinase assay GGAGCTCGGTACCCTCGAGGGATCCATGGCCATTTCGAAAGC TTAAGCAGAGATTACCTATCTAGATTAAGGGCATTTACGTTTGT StTCTP qPCR TCAAGATCTCCTCACCGGTG CATCTTCACCTCCACCCTCA StTCTP Yeast one-kybrid CGACGTACCAGATTACGCTCATATGTTGGTTTATCAAGATCTCCT CGATGCCCACCCGGGTGGAATTCTAGCACTTGATCTCCTTCAAG StTCTP luciferase complementary CGCGGTGGCGGCCGCTCTAGAATGTTGGTTTATCAAGATCTCCT TCAGCGTACCGAATTGGTACCCTAGCACTTGATCTCCTTCAAG StSnRK2.2 qPCR TTATGGAGTACGCAGCAGGT CCCACAGTCGACTTTGGTTG StSnRK2.3 qPCR TGTATGTCATGCTGGTGGGT ACGTTGATCTGGCTCCTCAA StSnRK2.4 qPCR TTTTGGCCTCGAATGCAACA ACGTTGATCTGGCTCCTCAA StSnRK2.6 qPCR GATCGCATCTGTCAAGCTGG ACTTGGGACGTGAATGCAAC ⑦Teacher, your suggestion is very good. We have conducted extensive research using StSN2 transgenic material and have published multiple research papers. 《The Cysteine-Rich Peptide Snakin-2 Negatively Regulates Tubers Sprouting through Modulating Lignin Biosynthesis and H2O2 Accumulation in Potato ", 《StSN2 interacts with the brassinosteroid signaling suppressor StBIN2 to maintain tuber dormancy》,《Snakin-2 interacts with cytosolic glyceraldehyde-3-phosphate dehydrogenase 1 to inhibit sprout growth in potato tubers》。 Therefore, there is no problem with the authenticity of the material. Potatoes reproduce asexually. We will place the harvested test tube potatoes back on MS medium for growth. Before placement, we need to perform StSN quantitative testing to meet our experimental needs. The seedlings grown from genetically modified tubers are our experimental materials, so we did not verify the leaves separately. I don't know if my explanation has dispelled your concerns, and if you are satisfied with this result. All authors have read and approved the resubmission of the manuscript! If you have any questions, please let me know! Thank you for your consideration of our paper and we are looking forward to hearing from you! Sincerely yours Shifeng Liu
Reviewer 2 Report
Comments and Suggestions for Authors
The manuscript “StTCTP positively regulates StSN2 to enhance drought stress tolerance in potato by scavenging reactive oxygen species” provides very interesting results. Overall, the manuscript is very well written, organized, and implicate a detailed analysis and discussion of many results.
However, I think the authors could clarify and improve few details. The main aspects to clarify is the physiological meaning of MDA content and Fv/Fm values, and the details includes, for example, the tobacco species used (different species are referred in different sections), or the absence of italic in the species names, as presented below:
Line 54 – meaning of GASA abbreviation must be presented, otherwise it is more difficult to understand the relation with gibberellin stimulates-like 2 (StSN2)
Line 63 – meaning of GAPC abbreviation must be presented
Line 57 – should be …Solanum tuberosum L. gibberellin stimulates-like 2 (StSN2)
Lines 69 to 71 – I think these sentences, referring the relation between mercury and the translation controlled tumour protein, are not relevant to the manuscript despite its refer to an abiotic stress.
Figure 1 A – should be “Drought stress for … “ instead of “Drought stress fore …”
Figure 1 B – should be “Days after … “ instead of “Dayd after…”
Line 135 – May the author explain what they pretend to say when they refer “PEG-6000 is a commonly used penetrating agent”?
As far as I know, PEG-6000 cannot penetrate in the plants, as can happen with PEG-4000. PEG-6000 can limit the soil water available to the be uptake by the plant because it increases the osmolarity.
Figure 2B- Figure 2 B – In OE27 with 5% PEG6000 the lowercase letter shouldn’t be a instead of b? OE11 and OE27 seems similar but have different lowercase letters. Also, OE-27 and WT are certainly different and have the same lowercase letter (b).
Line 166 – Here the authors refer that “peroxidase is the main intracellular reactive oxygen species (ROS) scavenger”, while after they consider catalase is the main enzyme involved in ROS detoxification. (line 331). This must be clarified.
Line 192- It is used Nicotiana benthamiana or Nicotiana tabacum, as mencioned in Material and Methods (line 376, )?
Line 231 – In Figure 5 – “The expression levels of the StTCTP gene“ could be “The expression levels of the StTCTP gene in potato“
Line 239 – should be “…PEG6000…, instead of PEG (only)
Line 246 – I think it is important to clarify the physiological meaning of Fv/Fm, a chlorophyll a fluorescence parameter, as an indicator of stress. Also, the range values of Fv/Fm for most plants should be given.
Line 248 – I think the sentence “…Fv/Fm values decreased for all leaves under stress…” is not totally correct. In OE-T8 the values seem similar at 0 days and 3 days after drought.
Figure 6B – It is important to refer the chlorophyll a fluorescence parameter presented in the images (probably Fv/Fm) and to include the colour legend.
Line 252 – “ RWC of OE-StTCTP was higher than WT, but water loss was lower …” or “RWC of OE-StTCTP was higher than WT, and water loss was lower…”
Line 262 – What is the physiological meaning of MDA accumulation? I think it is important to refer here and in discussion line 292.
Lines 317 to 319 – I think is not correct to refer “… I …” because the manuscript have several co-authors. The sentence shall be rewritten.
Line 319 e 320 – “… the water loss of the leaves and increases the RWC of the leaves…” should be rewritten to highlight the relation between water loss and RWC in leaves.
Line 376 – It is used Nicotiana tabacum, as referred here, or N. benthamiana, as referred in line 192? Also, italic is missing.
Line 381 – Are the seedlings grown only with coconut bran for 40 days? Which are the “standard conditions” used?
Line 382 – “Natural drought stress “ is withholding irrigation?
Line 390 – “…were planted in flower pots” with soil or coconut bran?
Line 390 – Sentence “The RWC assay, the functional leaves were measured the FW” must be rewritten.
Lines 402 to 409 – I think it must be explained what is quantified be the different approaches (e.g. enzyme activities or content, etc)
Lines 467 to 474 are repeated
Line 485 – Should be “ Measurement of chlorophyll a fluorescence” instead of “Measurement of chlorophyll fluorescence”

Author Response
Dear teacher After receiving your final response, I offer my sincerest apologies for the errors present in the manuscript. I have carefully reviewed your feedback and am in awe of your meticulous scientific approach. Moreover, I am grateful for the opportunity you have given me to resubmit my manuscript. Now we are submitting the revised manuscript entitled “StTCTP positively regulates StSN2 to enhance drought stress tolerance in potato by scavenging reactive oxygen species” (ID: ijms-3515110) for consideration for publication in《International Journal of Molecular Sciences》. In the revision process, we have implemented nearly all the suggestions and addressed all the comments made by the reviewers. We appreciate the time and effort that you and the reviewers dedicated to providing feedback on our manuscript. We are grateful for the insightful comments and valuable improvements to our paper. Those comments are valuable for revising and improving our paper with important guiding significance.We have made corrections according to the comments. The revised portions are marked in yellow on the paper. All page numbers refer to the revised manuscript file with tracked changes. I hope that the changes I’ve made resolve all your concerns about the article. I’m more than happy to make any further changes that will improve the paper and/or facilitate successful publication. Reviewer 1 Line 54 – meaning of GASA abbreviation must be presented, otherwise it is more difficult to understand the relation with gibberellin stimulates-like 2 (StSN2) Line 63 – meaning of GAPC abbreviation must be presented Line 57 – should be …Solanum tuberosum L. gibberellin stimulates-like 2 (StSN2) Lines 69 to 71 – I think these sentences, referring the relation between mercury and the translation controlled tumour protein, are not relevant to the manuscript despite its refer to an abiotic stress. Figure 1 A – should be “Drought stress for … “ instead of “Drought stress fore …” Figure 1 B – should be “Days after … “ instead of “Dayd after…” Line 135 – May the author explain what they pretend to say when they refer “PEG-6000 is a commonly used penetrating agent”? As far as I know, PEG-6000 cannot penetrate in the plants, as can happen with PEG-4000. PEG-6000 can limit the soil water available to the be uptake by the plant because it increases the osmolarity. Figure 2B- Figure 2 B – In OE27 with 5% PEG6000 the lowercase letter shouldn’t be a instead of b? OE11 and OE27 seems similar but have different lowercase letters. Also, OE-27 and WT are certainly different and have the same lowercase letter (b). Line 166 – Here the authors refer that “peroxidase is the main intracellular reactive oxygen species (ROS) scavenger”, while after they consider catalase is the main enzyme involved in ROS detoxification. (line 331). This must be clarified. Line 192- It is used Nicotiana benthamiana or Nicotiana tabacum, as mencioned in Material and Methods (line 376, )? Line 231 – In Figure 5 – “The expression levels of the StTCTP gene“ could be “The expression levels of the StTCTP gene in potato“ Line 239 – should be “…PEG6000…, instead of PEG (only) Line 246 – I think it is important to clarify the physiological meaning of Fv/Fm, a chlorophyll a fluorescence parameter, as an indicator of stress. Also, the range values of Fv/Fm for most plants should be given. Line 248 – I think the sentence “…Fv/Fm values decreased for all leaves under stress…” is not totally correct. In OE-T8 the values seem similar at 0 days and 3 days after drought. Figure 6B – It is important to refer the chlorophyll a fluorescence parameter presented in the images (probably Fv/Fm) and to include the colour legend. Line 252 – “ RWC of OE-StTCTP was higher than WT, but water loss was lower …” or “RWC of OE-StTCTP was higher than WT, and water loss was lower…” Line 262 – What is the physiological meaning of MDA accumulation? I think it is important to refer here and in discussion line 292. Lines 317 to 319 – I think is not correct to refer “… I …” because the manuscript have several co-authors. The sentence shall be rewritten. Line 319 e 320 – “… the water loss of the leaves and increases the RWC of the leaves…” should be rewritten to highlight the relation between water loss and RWC in leaves. Line 376 – It is used Nicotiana tabacum, as referred here, or N. benthamiana, as referred in line 192? Also, italic is missing. Line 381 – Are the seedlings grown only with coconut bran for 40 days? Which are the “standard conditions” used? Line 382 – “Natural drought stress “ is withholding irrigation? Line 390 – “…were planted in flower pots” with soil or coconut bran? Line 390 – Sentence “The RWC assay, the functional leaves were measured the FW” must be rewritten. Lines 402 to 409 – I think it must be explained what is quantified be the different approaches (e.g. enzyme activities or content, etc) Lines 467 to 474 are repeated Line 485 – Should be “ Measurement of chlorophyll a fluorescence” instead of “Measurement of chlorophyll fluorescence” Comment 1: ①Line 54 – meaning of GASA abbreviation must be presented, otherwise it is more difficult to understand the relation with gibberellin stimulates-like 2 (StSN2) ②Line 63 – meaning of GAPC abbreviation must be presented ③Line 57 – should be …Solanum tuberosum L. gibberellin stimulates-like 2 (StSN2) Response 1: Thank you for your suggestion. I made the following modifications: ①I have revised the abstract as follows (page1 line13-35): Abstract: Drought is a negative agronomic effect that can lead to an increase in reactive oxygen species (ROS) levels. Excessive can severely alter cell membrane fluidity and permeability, significantly reducing cell viability. Gibberellic acid-stimulated Arabidopsis (Snakin/GASA) gene family plays an important role as antioxidants in inhibiting the accumulation of ROS and improving crop drought resistance. However, the regulatory mechanism of potato StSnakin-2 (StSN2) in response to drought, as well as how StSN2 expression is regulated, is not well understood. In this study, we found that StSN2 was induced by drought. Overexpression of StSN2 significantly increased drought tolerance, whereas silencing StSN2 increased sensitivity to drought. Overexpression of StSN2 resulted in higher antioxidant enzyme (superoxide dismutase (SOD), catalase (CAT), and peroxidase (POD)) activity, and lowered hydrogen peroxide (H2O2) and malondialdehyde (MDA) accumulation during drought stress. Also, overexpression of StSN2 increased relative water content of leaves (RWC) and reduces water loss in leaves. We screened the upstream regulatory protein translation controlled tumour protein (StTCTP) of StSN2 through DNA pull-down combined with mass spectrometry. Yeast one-hybrid (YIH), electrophoretic mobility shift assays (EMSA) and luciferase reporting assay (LUC) indicated that StTCTP binds the StSN2 promoter. Like StSN2, StTCTP was highly expressed in response to drought. Overexpression of StTCTP increased the photosynthetic rate and CAT enzyme activity, and lowered H2O2 and MDA accumulation during drought. Meanwhile, overexpression of StTCTP increased leaf RWC and reduced water loss. Our research strongly suggested that StSN2 effectively cleared ROS and significantly boosted the drought resistance of potatoes. Furthermore, as a transcriptional activator of StSN2, StTCTP, much like StSN2, also enhanced the potato's drought tolerance. The result provided a foundation for further study of StSN2 regulatory mechanisms under drought stress. ②I have added the full name of GAPC in the manuscript (page 2, line69-70). ③ I didn't write this sentence clearly, which caused confusion for the teacher. This is a study on the Snakin-2 gene in Petunia hybrida. I have rewritten this sentence.Overexpression of GIP2 (Snakin-2) in Petunia hybrida led to reduced accumulation of H2O2 in leaves following wounding.(page 2, line62-63) Comment 2: ①Lines 69 to 71 – I think these sentences, referring the relation between mercury and the translation controlled tumour protein, are not relevant to the manuscript despite its refer to an abiotic stress. ②Figure 1 A – should be “Drought stress for … “ instead of “Drought stress fore …” ③Figure 1 B – should be “Days after … “ instead of “Dayd after…” Response 2 :We apologize for the issues in the manuscript. I made the following modifications: ①Based on the reviewer's suggestion, I deleted the sentence. ②-③ Figure 1. The effects of drought stress on WT potatoes and the expression levels of StSN2 gene at the same time. (A) Growth status of potatoes under drought treatment. Scale bar =10 cm. (B) The expression level of the StSN2 gene under drought treatment. (C) The spatial expression of potato StSN2 on the 12th day of drought treatment. Data are means±SD of three biological replicates. Different small letters represent significant differences (p < 0.05). Comment 3: ①Line 135 – May the author explain what they pretend to say when they refer “PEG-6000 is a commonly used penetrating agent”? As far as I know, PEG-6000 cannot penetrate in the plants, as can happen with PEG-4000. PEG-6000 can limit the soil water available to the be uptake by the plant because it increases the osmolarity. ②Figure 2B- Figure 2 B – In OE27 with 5% PEG6000 the lowercase letter shouldn’t be a instead of b? OE11 and OE27 seems similar but have different lowercase letters. Also, OE-27 and WT are certainly different and have the same lowercase letter (b). Response 3 : We are immensely grateful to the reviewers for bringing this issue to our attention. I made the following modifications: ① I am very sorry for the misunderstanding caused by my unclear expression to the teacher. We used PEG-6000 to simulate plant drought stress by reducing the absorption of water by plants. I have rewritten the sentence. PEG-6000 is a frequently used osmotic agent capable of restricting plants' absorption of water from soil, thereby simulating the drought stress (page 3, line 133-134). ②I reanalyzed the data and indeed, as the teacher said, I redrawn the graph. The result is shown in the figure: Figure 2. Overexpression StSN2 enhances tolerance to drought stress in potato. (A) Images of transgenic OE-StSN2, RNAi-StSN2, and WT potato lines after four weeks of growth on MS medium in normal conditions or 5% PEG-6000. Scale bar=2 cm. (B) Quantification of plant length. (C) Quantification of root length. (D) Rates of water loss in detached leaves of seedlings from OE-StSN2, RNAi-StSN2, and WT were measured every 0.5 h over a total of 2.5 h. (E) Leaf RWC of OE-StSN2, RNAi-StSN2, and WT potato plants. Data are means±SD of three biological replicates. Different lowercase letters indicate significant differences for the same treatment (P≤0.05). Comment 4: ①Line 166 – Here the authors refer that “peroxidase is the main intracellular reactive oxygen species (ROS) scavenger”, while after they consider catalase is the main enzyme involved in ROS detoxification. (line 331). This must be clarified. Line 192- It is used Nicotiana benthamiana or Nicotiana tabacum, as mencioned in Material and Methods (line 376, )? ②Line 231 – In Figure 5 – “The expression levels of the StTCTP gene“ could be “The expression levels of the StTCTP gene in potato“ ③Line 239 – should be “…PEG6000…, instead of PEG (only) ④Line 246 – I think it is important to clarify the physiological meaning of Fv/Fm, a chlorophyll a fluorescence parameter, as an indicator of stress. Also, the range values of Fv/Fm for most plants should be given. Response 4 : I am extremely grateful to the reviewer for raising this question. I made the following modifications: ①What we want to express is that antioxidant enzymes are the main ROS scavengers, but my expression is incorrect. CAT is a type of antioxidant enzyme that can also clear ROS. We have made modifications to this: Antioxidant enzymes are the main intracellular ROS scavenger, which can effectively neutralize and reduce the damage caused by oxidative stress to cells[32] (page 4, line 165-166). ② In the materials and methods, we used Nicotiana benthamiana L, which has been modified (page 10, line 374). ③I have made additions and modifications, PEG-6000 has been supplemented (page 7, line 243). ④The teacher's suggestion is very good, and we have consulted relevant materials. Generally speaking, the Fv/Fm value of normal plants ranges between 0.75 and 0.85. (page 7, line 253). Comment 5: ①Figure 6B – It is important to refer the chlorophyll a fluorescence parameter presented in the images (probably Fv/Fm) and to include the colour legend. ②Line 252 – “ RWC of OE-StTCTP was higher than WT, but water loss was lower …” or “RWC of OE-StTCTP was higher than WT, and water loss was lower…” ③Line 262 – What is the physiological meaning of MDA accumulation? I think it is important to refer here and in discussion line 292. ④Lines 317 to 319 – I think is not correct to refer “… I …” because the manuscript have several co-authors. The sentence shall be rewritten. ⑤Line 319 e 320 – “… the water loss of the leaves and increases the RWC of the leaves…” should be rewritten to highlight the relation between water loss and RWC in leaves. Response 5 : I am extremely grateful to the reviewer for raising this question. I made the following modifications: ①The suggestions given by the teachers are very good. Figure 6B is just the image we took, and the specific Fv/Fm values are reflected in Figure 6C, which I cannot describe clearly. I have rewritten this paragraph. Our research founded that as drought increases, the degree of wilting in plants intensifies; however, the growth of OE-T1 and OE-T8 is superior to that of WT (Figure 6A). The fluorescence results of the leaves were consistent with the degree of wilting observed in potatoes (Figure 6B) (page 7, line 248-251). ②I made the modifications according to the teacher's suggestion. RWC of OE-StTCTP was higher than WT, and water loss was lower (page 7, line 259-260). ③ MDA is the final product of lipid peroxidation reaction, and its level can be an important indicator for evaluating the degree of cell membrane damage (page 9, line 302-303). ④I made the modifications according to the teacher's suggestion. In our previous research, we also observed (page 9, line 315). ⑤ I made the modifications according to the teacher's suggestion. The water loss and relative water content of detached leaves are usually negatively correlated. StSN2 effectively enhanced potato tolerance to drought by reducing leaf water loss and increasing leaf relative water content (page 9, line 317-319). Comment 6: ①Line 376 – It is used Nicotiana tabacum, as referred here, or N. benthamiana, as referred in line 192? Also, italic is missing. ②Line 381 – Are the seedlings grown only with coconut bran for 40 days? Which are the “standard conditions” used? ③Line 382 – “Natural drought stress “ is withholding irrigation? ④Line 390 – “…were planted in flower pots” with soil or coconut bran? ⑤Line 390 – Sentence “The RWC assay, the functional leaves were measured the FW” must be rewritten. ⑥Lines 402 to 409 – I think it must be explained what is quantified be the different approaches (e.g. enzyme activities or content, etc) ⑦Lines 467 to 474 are repeated ⑧Line 485 – Should be “ Measurement of chlorophyll a fluorescence” instead of “Measurement of chlorophyll fluorescence Response 6 : I am extremely grateful to the reviewer for raising this question. I made the following modifications: ①Our entire experiment used Nicotiana benthamiana, and I have made modifications in the Materials and Methods section (page 10, line 374). Thank you. ②I made the modifications according to the teacher's suggestion. We transferred WT potato tissue culture seedlings with a length of 6-8 cm into pots filled with coconut bran and cultivated them in a growth chamber with a temperature of 20℃ and a 16 h/8 h light and dark cycle for 60 days (page 10, line 378-380). ③Normal drought is treated without watering ④I made the modifications according to the teacher's suggestion. StTCTP transgenic and WT tissue cultured seedlings were planted in flower pots with coconut bran (page 10, line 389). ⑤ I made the modifications according to the teacher's suggestion. For the RWC assay, we measured the FW of the functional leaves (page 11, line 398-399). ⑥I made the modifications according to the teacher's suggestion(page 11, line 403-413). Staining with DAB was performed using a previously described method[46]. First, the leaves were thoroughly cleaned, then immersed in DAB staining solution and subjected to overnight dark treatment. Next, the leaves were bleached using alcohol. Finally, photographs of the bleached leaves were taken for observation. All samples were prepared for enzyme activity by homogenizing 0.1 g of leaves in a solution of 0.01 mM pH 7.2 phosphate buffer saline. The homogenate was centrifuged at 12 000 rpm for 10 min at 4°C. The activities of SOD, CAT and POD were measured separately by using a SOD assay kit (Cat. BC0175), CAT assay kit (Cat. BC0205), POD assay kit (Cat. BC0095) produced by Solarbio life science. The levels of MDA and H2O2 were performed based on the procedure described in the manufacturer’s directions (Solarbio, China)[47-49]. ⑦This was my mistake, I have already deleted the content. ⑧I made the modifications according to the teacher's suggestion(page 12, line 483-488).. 4.13. Measurement of chlorophyll a fluorescence Before the measurement, the potato seedlings need to be placed in dark conditions for 30 minutes. Chlorophyll a fluorescence was recorded following 1 second of light exposure. The potential photosynthetic efficiency (FV/FM) was recorded by taking the variable fluorescence (FV) and dividing it by the maximal fluorescence (FM)[56]. Chlorophyll a fluorescence was measured using an IMAGING-PAM-MAXI chlorophyll fluorescence imaging system (Heinz Walz GmbH, Effectrich, Germany) [57]. All authors have read and approved the resubmission of the manuscript! If you have any questions, please let me know! Thank you for your consideration of our paper and we are looking forward to hearing from you! Sincerely yours Shifeng Liu
Reviewer 3 Report
Comments and Suggestions for Authors
In the manuscript: “StTCTP positively regulates StSN2 to enhance drought stress 2 tolerance in potato by scavenging reactive oxygen species”, author found that StSN2 and StTCTP was induced by drought. StSN2 overexpression and silencing lines confirmed StSN2 positively regulated drought tolerance, and StTCTP is the upstream regulatory protein of StSN2 through DNA pull-down combined with mass spectrometry. overexpression of StSN2 or StTCTP had higher antioxidant enzyme activity, and lowered H2O2 and MDA accumulation, indicating that they play the important role of the ROS system during drought stress. There are some issues as following:
- “the expression 121 of StSN2 did not change” in line 121-122, whereas, Figure 1B showed significant differences (d c), but the two are inconsistent.
- The legends of Figure 4 lacked descriptions of D and E.
- The legends of “Different lowercase letters indicate significant differences” in Figures 2, 3, and 7 must added for the same treatment, that is to say “Different lowercase letters indicate significant differences for the same treatment”
- Figure 4 has an error in the citation in lines 194-199 of the text. Such as Figure 4D or 4E does not appear in the text, and Figure 4C is not placed in the right place.
- Reference 41 and Supplementary File. 1 does not appear in the text.
- Material and method:
1) How the image of Figure 6B was obtained, the description is not detailed or omitted.
2) 4.11 and 4.12 are the same.
- There are also many problems in references.
- Volumes, pages or article number are missing in reference 1、21 and etc.
- The symbols between the page numbers should be consistent, such as “–” of reference 15, “-” of reference 2
- The year of publication is in bold or not, for example, The year of publication in most of references are bold, whereas reference 23、31 and 33 are not bold.
- The symbol before the page should be consistent, comma or colon.
- Page numbers must be fully written, for example, in reference “172-88” changed into “172-188”.
- No citation of reference 41 appears in the text.
- Some formatting issues:
1) in line 20 “(SOD, CAT, and POD)activity” changed into “(SOD, CAT, and POD) activity”;
2) in line 35 “cells [1] It is” changed into “cells [1]. It is”;
3) in line 40 “(Solanum tuberosum L.)is the four” changed into “(Solanum tuberosum L.) is the four”;
4) in line 120 “Figure 1A). ” changed into “Figure 1A). ”;
5) in Supplementary Table 1, there is a missing horizontal line below the table.
6) in line 187 “expression. (Supplementary Figure S2)” changed into “expression (Supplementary Figure S2)”

The English could be improved to more clearly express the research
Author Response
Dear teacher After receiving your final response, I offer my sincerest apologies for the errors present in the manuscript. I have carefully reviewed your feedback and am in awe of your meticulous scientific approach. Moreover, I am grateful for the opportunity you have given me to resubmit my manuscript. Now we are submitting the revised manuscript entitled “StTCTP positively regulates StSN2 to enhance drought stress tolerance in potato by scavenging reactive oxygen species” (ID: ijms-3515110) for consideration for publication in《International Journal of Molecular Sciences》. In the revision process, we have implemented nearly all the suggestions and addressed all the comments made by the reviewers. We appreciate the time and effort that you and the reviewers dedicated to providing feedback on our manuscript. We are grateful for the insightful comments and valuable improvements to our paper. Those comments are valuable for revising and improving our paper with important guiding significance.We have made corrections according to the comments. The revised portions are marked in yellow on the paper. All page numbers refer to the revised manuscript file with tracked changes. I hope that the changes I’ve made resolve all your concerns about the article. I’m more than happy to make any further changes that will improve the paper and/or facilitate successful publication. Reviewer 3 “the expression 121 of StSN2 did not change” in line 121-122, whereas, Figure 1B showed significant differences (d c), but the two are inconsistent. The legends of Figure 4 lacked descriptions of D and E. The legends of “Different lowercase letters indicate significant differences” in Figures 2, 3, and 7 must added for the same treatment, that is to say “Different lowercase letters indicate significant differences for the same treatment” Figure 4 has an error in the citation in lines 194-199 of the text. Such as Figure 4D or 4E does not appear in the text, and Figure 4C is not placed in the right place. 解决 Reference 41 and Supplementary File. 1 does not appear in the text. Material and method: How the image of Figure 6B was obtained, the description is not detailed or omitted. 2) 4.11 and 4.12 are the same. 1.There are also many problems in references. Volumes, pages or article number are missing in reference 1、21 and etc. The symbols between the page numbers should be consistent, such as “–” of reference 15, “-” of reference 2 The year of publication is in bold or not, for example, The year of publication in most of references are bold, whereas reference 23、31 and 33 are not bold. The symbol before the page should be consistent, comma or colon. Page numbers must be fully written, for example, in reference “172-88” changed into “172-188”. No citation of reference 41 appears in the text. Some formatting issues: 1)in line 20 “(SOD, CAT, and POD)activity” changed into “(SOD, CAT, and POD) activity”; 2)2) in line 35 “cells [1] It is” changed into “cells [1]. It is”;缺少了一个句号,我已经补充。 3) in line 40 “(Solanum tuberosum L.)is the four” changed into “(Solanum tuberosum L.) is the four”; 4) in line 120 “Figure 1A). ” changed into “Figure 1A). ”; 5) in Supplementary Table 1, there is a missing horizontal line below the table. 6) in line 187 “expression. (Supplementary Figure S2)” changed into “expression (Supplementary Figure S2)” Comment 1: ①“the expression 121 of StSN2 did not change” in line 121-122, whereas, Figure 1B showed significant differences (d c), but the two are inconsistent. ②The legends of Figure 4 lacked descriptions of D and E. ③The legends of “Different lowercase letters indicate significant differences” in Figures 2, 3, and 7 must added for the same treatment, that is to say “Different lowercase letters indicate significant differences for the same treatment” ④Figure 4 has an error in the citation in lines 194-199 of the text. Such as Figure 4D or 4E does not appear in the text, and Figure 4C is not placed in the right place. ⑤Reference 41 and Supplementary File. 1 does not appear in the text. Response 1: Thank you for the teachers' suggestions. I made the following modifications: ①I have rewritten the content of this paragraph(Page 3,line 119-122). As follows: On the 0th and the 4th days, the expression of StSN2 increased rapidly. However, on the 8th and 12th days, the expression of StSN2 increased even more significantly, reaching levels 6.5 times and 12.8 times higher than on day 0, respectively (Figure 1B). ②I have rewritten the paragraph and placed the image in the appropriate location.(Page 5-6,line 189-201). The interaction between the StSN2 promoter and StTCTP was confirmed by LUC (Figure 2A-C), EMSA (Figure 4D), and YIH (Figure 4E).We validated this interaction using a dual luciferase assay, where the StSN2 promoter was placed upstream of luciferase (Figure 4A) and coexpressed with StTCTP in N. benthamiana leaves (Figure 4C). The results of LUC showed that as the concentration of StTCTP increased, the interaction strength between proStSN2 and StTCTP continued to strengthen, and the LUC activity increased by 1.67 to 2.8 times relative to the empty effect carrier (Figure 4B). The results of the EMSA indicate that when the His protein alone binds to the probe, the electrophoresis result displays a single band. However, when the StTCTP-His protein binds to the probe, a shifted band is observed. As the concentration of the probe increases, the retardation effect gradually intensifies until the free probe completely disappears. Notably, when unlabeled competitive probes are added and their amount exceeds that of the labeled probes, the retardation effect is reversed, leading to disappearance of the shifted band (Figure 4D). To confirm StTCTP exhibits transcriptional activation activity, we constructed pHIS-proStSN2 and pGADT7-StTCTP vectors, both the constructs were transformed into yeast cells. Positive clones grew on SD medium lacking Trp, His, Leu and containing 3-AT. However, proStSN2 + AD-empty was unable to grow under the same condition (Figure 4E). These data indicate that StTCTP functions as a transcriptional activator of StSN2. Figure 4. StTCTP and StSN2 promoter interaction verification. (A-C) Dual-luciferase assays (D) electrophoretic mobility shift assay. (E) Yeast one-hybrid assay. Data are means±SD of three biological replicates. Different lowercase letters indicate significant differences for the same treatment (P≤0.05). ③I have added the sentence 'Different lower case letters indicate significant differences for the same treatment' in the captions of images 2, 3, and 7. ④ I have already provided a response in ②, please refer to it, teacher. ⑤Reference 41 and Supplementary File. 1 in the manuscript have been deleted. Comment 2: ①How the image of Figure 6B was obtained, the description is not detailed or omitted. ②2) 4.11 and 4.12 are the same. ③1.There are also many problems in references. Volumes, pages or article number are missing in reference 1、21 and etc. The symbols between the page numbers should be consistent, such as “–” of reference 15, “-” of reference 2 The year of publication is in bold or not, for example, The year of publication in most of references are bold, whereas reference 23、31 and 33 are not bold. The symbol before the page should be consistent, comma or colon. Page numbers must be fully written, for example, in reference “172-88” changed into “172-188”. No citation of reference 41 appears in the text. Response 2 :We apologize for the issues in the manuscript. I made the following modifications: ① Figure 6B depicts chlorophyll a fluorescence imaging of potato leaves under drought stress, as described in Materials and Methods 4.13 (page 13 line 486-491). Please consult the teacher. ②4.11 and 4.12 are the same, I have already deleted one. ③I carefully checked the references in the manuscript and found that some of them did have issues. I have made corrections to the references again. Reference 41 has also been deleted. Comment 3: ①1) in line 20 “(SOD, CAT, and POD)activity” changed into “(SOD, CAT, and POD) activity”; ②2) in line 35 “cells[1]It is” changed into “cells[1]. It is”; ③3) in line 40 “(Solanum tuberosumL.)is the four” changed into “(Solanum tuberosumL.) is the four” Response 3 : We are immensely grateful to the reviewers for bringing this issue to our attention. I made the following modifications: ① I have revised it as follows: (superoxide dismutase (SOD), catalase (CAT), and peroxidase (POD)) activity. (Page 1, line21-22) ②I have revised it as follows:Water is the source of life and one of the main components of cells. (page1, line 39) ③I have revised it as follows: Potato (Solanum tuberosum L.) is the fourth food on earth ( page1, line 44). Comment 3: ①4) in line 120 “Figure 1A). ” changed into “Figure 1A). ”; ②5) in Supplementary Table 1, there is a missing horizontal line below the table. ③6) in line 187 “expression. (Supplementary Figure S2)” changed into “expression (Supplementary Figure S2)”。 Response 3 :We are immensely grateful to the reviewers for bringing this issue to our attention. I made the following modifications: ① I have revised it as follows: value (Figure 1A). (page3, line 117) ② I have revised it as follows: Supplementary Table 1. Quantitative RT-PCR for RNA sequencing validation. Gene Purpose Forward/Reverse StSN2 qPCR TAACAGATGTAGCCACTGAC ACAACAAGTTCCACATGCCC Pro-StSN2 Yeast one-kybrid CGACTCACTATAGGGCGAATTCTGTAGTTGAACTTTTTATCA GACCGCGGATCGATTCGCGAACGCGTTGGAATTTGAAATATTTTTCT StSN2 Luciferase complementary CTATAGGGCGAATTGGGTACCTGTAGTTGAACTTTTTATCA TAGAACTAGTGGATCCCCCGGGTGGAATTTGAAATATTTTTC StSN2 Kinase assay GGAGCTCGGTACCCTCGAGGGATCCATGGCCATTTCGAAAGC TTAAGCAGAGATTACCTATCTAGATTAAGGGCATTTACGTTTGT StTCTP qPCR TCAAGATCTCCTCACCGGTG CATCTTCACCTCCACCCTCA StTCTP Yeast one-kybrid CGACGTACCAGATTACGCTCATATGTTGGTTTATCAAGATCTCCT CGATGCCCACCCGGGTGGAATTCTAGCACTTGATCTCCTTCAAG StTCTP luciferase complementary CGCGGTGGCGGCCGCTCTAGAATGTTGGTTTATCAAGATCTCCT TCAGCGTACCGAATTGGTACCCTAGCACTTGATCTCCTTCAAG StSnRK2.2 qPCR TTATGGAGTACGCAGCAGGT CCCACAGTCGACTTTGGTTG StSnRK2.3 qPCR TGTATGTCATGCTGGTGGGT ACGTTGATCTGGCTCCTCAA StSnRK2.4 qPCR TTTTGGCCTCGAATGCAACA ACGTTGATCTGGCTCCTCAA StSnRK2.6 qPCR GATCGCATCTGTCAAGCTGG ACTTGGGACGTGAATGCAAC ③ I have revised it as follows: GUS gene expression (Supplementary Figure S2). (page5, line 187) All authors have read and approved the resubmission of the manuscript! If you have any questions, please let me know! Thank you for your consideration of our paper and we are looking forward to hearing from you! Sincerely yours Shifeng Liu
Reviewer 4 Report
Comments and Suggestions for Authors
Potato is the fourth crop plant, while abiotic stress such as drought stress is a major stress factor that affects potato plant growth, development, and yield. This manuscript reported that StTCTP positively regulates StSN2 to enhance drought stress tolerance in potato by scavenging reactive oxygen species. It is important to understand the molecular mechanisms of drought tolerance. In addition, the manuscript was well designed and wrote, and obtained expected results. However, I think the following comments should be considered.
(1) ABSTRACT: OK.
(2) INTRODUCTION: OK.
(3) RESULTS: First, the tense of the subtitles should be consistent, such as 2.1 and 2.4 are past tense, while others present tense. Second, in general, drought stress causes osmotic stress, namely celllular dehydration, as shown in figure 1, so some parameters in figures 2, 3, 7 were expressed in FW, which is incorrect, they should be expressed in DW.
(4) DISCUSSION: Besides purpose, the main findings should be presented in the first paragraph and then further discussed in other paragraphs.
(5) METHODS: In 4.2, some parameters such as temperature, relative humidity should be supplemented. Also, why 5%PEG was used to simulate drought stress should be explain.
(6) LANGUAGE: Language should be carefully polished, such as arabic number can not be used at the beginning of a sentence.
Comments on the Quality of English Languageabove
Author Response
Dear teacher After receiving your final response, I offer my sincerest apologies for the errors present in the manuscript. I have carefully reviewed your feedback and am in awe of your meticulous scientific approach. Moreover, I am grateful for the opportunity you have given me to resubmit my manuscript. Now we are submitting the revised manuscript entitled “StTCTP positively regulates StSN2 to enhance drought stress tolerance in potato by scavenging reactive oxygen species” (ID: ijms-3515110) for consideration for publication in《International Journal of Molecular Sciences》. In the revision process, we have implemented nearly all the suggestions and addressed all the comments made by the reviewers. We appreciate the time and effort that you and the reviewers dedicated to providing feedback on our manuscript. We are grateful for the insightful comments and valuable improvements to our paper. Those comments are valuable for revising and improving our paper with important guiding significance.We have made corrections according to the comments. The revised portions are marked in yellow on the paper. All page numbers refer to the revised manuscript file with tracked changes. I hope that the changes I’ve made resolve all your concerns about the article. I’m more than happy to make any further changes that will improve the paper and/or facilitate successful publication. Reviewer 4 RESULTS: First, the tense of the subtitles should be consistent, such as 2.1 and 2.4 are past tense, while others present tense. Second, in general, drought stress causes osmotic stress, namely celllular dehydration, as shown in figure 1, so some parameters in figures 2, 3, 7 were expressed in FW, which is incorrect, they should be expressed in DW. DISCUSSION: Besides purpose, the main findings should be presented in the first paragraph and then further discussed in other paragraphs. METHODS: In 4.2, some parameters such as temperature, relative humidity should be supplemented. Also, why 5%PEG was used to simulate drought stress should be explain. Comment 1: RESULTS: First, the tense of the subtitles should be consistent, such as 2.1 and 2.4 are past tense, while others present tense. Second, in general, drought stress causes osmotic stress, namely celllular dehydration, as shown in figure 1, so some parameters in figures 2, 3, 7 were expressed in FW, which is incorrect, they should be expressed in DW. Response 1: Thank you for your suggestion. The tense of the entire document is in the past tense. I have checked the entire document and ensured consistency in the tense throughout. I have also made modifications to figures 2, 3, 7. As follows: Figure 2. Overexpression StSN2 enhances tolerance to drought stress in potato. (A) Images of transgenic OE-StSN2, RNAi-StSN2, and WT potato lines after four weeks of growth on MS medium in normal conditions or 5% PEG-6000. Scale bar=2 cm. (B) Quantification of plant length. (C) Quantification of root length. (D) Rates of water loss in detached leaves of seedlings from OE-StSN2, RNAi-StSN2, and WT were measured every 0.5 h over a total of 2.5 h. (E) Leaf RWC of OE-StSN2, RNAi-StSN2, and WT potato plants. Data are means±SD of three biological replicates. Different lowercase letters indicate significant differences for the same treatment (P≤0.05). Figure 3. Overexpression of StSN2 in potato reduces ROS accumulation during drought. (A) OE-StSN2, RNAi-StSN2, and WT potato lines were grown on MS medium in normal conditions or 5% PEG-6000 for four weeks. A DAB staining. Scale bar = 1 cm. (B) Quantification of H2O2 accumulation. (C) SOD activity. (D) MDA accumulation. (E) POD activity (F) CAT activity. Data are means±SD of three biological replicates. Different lowercase letters indicate significant differences for the same treatment (P≤0.05). Figure 7. ROS accumulation and CAT activity in leaves of potatoes. (A) H2O2 content. (B) MDA content. (C) CAT enzyme activity. OE-StTCTP and WT potato lines were treated for 5 days with 5% PEG-6000. Data are means±SD of three biological replicates. Different lowercase letters indicate significant differences for the same treatment (P≤ 0.05). Comment 2:DISCUSSION: Besides purpose, the main findings should be presented in the first paragraph and then further discussed in other paragraphs. Response 2: Thank you for your suggestion. We have rewritten this paragraph(page8-9,line284-298). As follows: Drought inhibits cell growth, leading to a significant annual reduction in crop yield. Between 2006 and 2015, the average annual economic loss in China due to drought was approximately 12.8 billion US dollars (USD), which accounted for 0.16% of the country’s gross domestic product (GDP)[34]. Potatoes, the fourth-largest food crop globally, are highly affected by drought, but drought-resistant varieties are rare due to their narrow genetic background. Studying drought resistance mechanisms in potatoes is crucial for boosting yield and quality. Drought stress triggers ROS production in plants, disrupting the oxidant-antioxidant balance and causing oxidative stress[33, 35]. Research shows that cysteine sites in Snakin/GASA proteins form disulfide bonds vital for redox regulation[8]. Overexpression of related proteins like GIP2 in petunia and FsGASA4 in Arabidopsis reduces oxidative stress and enhances tolerance to various stresses[9, 36]. StSN2, an antibacterial peptide in potatoes belonging to the Snakin/GASA family[37], exhibits a specific expression pattern under drought stress, with its transcription level rising as stress intensifies (Figure 1B-C). Thus, StSN2 is a key drought-responsive gene in potatoes. Comment 3:METHODS: In 4.2, some parameters such as temperature, relative humidity should be supplemented. Also, why 5%PEG was used to simulate drought stress should be explain. Response 3: Thank you for your suggestion. We added relevant content in 4.2(page10,line378-380). As follows: We transferred WT potato tissue culture seedlings with a length of 6-8 cm into pots filled with coconut bran and cultivated them in a growth chamber with a temperature of 20℃ and a 16 h/8 h light and dark cycle for 60 days. For the selection of 5% PEG-6000 concentration, we consulted the relevant literature 《Transcriptome analysis of potato stem segments under PEG simulated srought stress》 DOI: 10.7606/j.issn.1000-4025.2020.03.0403;《Effects of PEG-6000 stress on potato tissue culture seedlings under ex vivo conditions》DOI: 10.19720/j.cnki. issn. 1005-9369.2015.10.001; 《Identification of drought tolerant potato varieties and bioinformatics analysis of StAREB gene》 DOI: 10.13271/j.mpb. 019.004896. In addition, I also conducted preliminary experiments and found that the concentration of 5% PEG-6000 had the best experimental effect, so I used this concentration. Can my explanation dispel the concerns of the teachers? Thank you again for your hard work. Comment 4: LANGUAGE: Language should be carefully polished, such as arabic number can not be used at the beginning of a sentence. Response 4: We apologize for our poor writings. Thank you for your suggestion! we once again invited a friend of us who is a native English speaker from the USA to help proofread this manuscript. We have carefully and thoroughly proofread the manuscript to correct all the grammar and spelling mistakes. And we hope the revised manuscript could be acceptable for you. All authors have read and approved the resubmission of the manuscript! If you have any questions, please let me know! Thank you for your consideration of our paper and we are looking forward to hearing from you! Sincerely yours Shifeng Liu
Round 2
Reviewer 1 Report
Comments and Suggestions for Authors
Thank you very much for your response to the questions I raised. I have two additional points to bring to your attention:
- Please provide a more comprehensive account of the author contributions. Presently, the description does not clarify who was responsible for conducting the experiments. Again, If the experiments weren't carried out by you alone but were a team effort, please use "WE" to accurately reflect the collaborative nature of the work.
- For the writing, there are multiple errors in the current “author contributions” section.
Author Response
Dear Ms. Siriratchakorn Sathiensathaporn
After receiving your reply, I sincerely apologize for the errors in the manuscript. I am very grateful that you have given me another opportunity to resubmit my manuscript. We are currently submitting a revised manuscript titled "StTCTP Actively Regulates StSN2 by Clearing Reactive Oxygen Species to Improve Potato Drought Resistance" (ID: ijms-3515110) for publication in the International Journal of Molecular Science.
During the revision process, we almost implemented all the suggestions and addressed all the comments raised by the reviewers. We appreciate the time and effort you and the reviewers have invested in providing feedback on our manuscript. We appreciate everyone's insightful comments and valuable improvements on our paper. These opinions have important guiding significance for revising and improving our paper. We have made corrections based on the feedback. The modified part is marked in yellow on the paper. All page numbers refer to the revised manuscript file and are accompanied by a record of changes. I hope the changes I have made can address all your concerns about this article. I am more than willing to make any further modifications to improve the paper and/or facilitate its successful publication.
Reviewer 1
Comments and Suggestions for Authors
Thank you very much for your response to the questions I raised. I have two additional points to bring to your attention:
Please provide a more comprehensive account of the author contributions. Presently, the description does not clarify who was responsible for conducting the experiments. Again, If the experiments weren't carried out by you alone but were a team effort, please use "WE" to accurately reflect the collaborative nature of the work.
For the writing, there are multiple errors in the current “author contributions” section.
Respond: I strongly agree with my teacher's suggestion and apologize for my unclear expression earlier. Based on my teacher's advice, I have rewritten the "Author Contributions" section as follows: (page13, line494-487)
S.L. drafted the manuscript, analyzed data, and prepared figures; F.Z. and H.F. provided materials and described results; Q.W. and X.W. reviewed and edited the manuscript; X.L. and L. Y. conceived and designed this study. All authors have read and agreed to the published the manuscript.
All authors have read and approved the resubmission of the manuscript! If you have any questions, please let me know!
Thank you for your consideration of our paper and we are looking forward to hearing from you!
Sincerely yours
Shifeng Liu

Reviewer 3 Report
Comments and Suggestions for Authors
no
Author Response
Dear Ms. Siriratchakorn Sathiensathaporn
I received your letter and I am very excited. I am very grateful for the opportunity to submit my manuscript again. I am very grateful for the hard work of the teacher, who has put in a lot of effort to revise my manuscript. Here, I would like to thank the teachers who reviewed it again. In future research, I will work hard and strive to do better in scientific research! Thank you very much! We are currently submitting a revised manuscript titled "StTCTP Actively Regulates StSN2 by Clearing Reactive Oxygen Species to Improve Potato Drought Resistance" (ID: ijms-3515110) for publication in the International Journal of Molecular Science.
All authors have read and approved the resubmission of the manuscript! If you have any questions, please let me know!
Thank you for your consideration of our paper and we are looking forward to hearing from you!
Sincerely yours
Shifeng Liu

Reviewer 4 Report
Comments and Suggestions for Authors
The manuscript has been improved according to the comments, so I think it should be considered to accept to publish.
Comments on the Quality of English Language
no
Author Response

(The authors gave the same response as above.)
